# Characterizing sources of high surface ozone events in the southwestern
# U.S. with intensive field measurements and two global models
Li Zhang[1,2], Meiyun Lin[1,2*], Andrew O. Langford[3], Larry W. Horowitz[2], Christoph J. Senff[3,4], Elizabeth
Klovenski[5], Yuxuan Wang[5], Raul J. Alvarez II[3], Irina Petropavlovskikh[4,6], Patrick Cullis[4,6], Chance W.
Sterling[4,6,7], Jeff Peischl[3,4], Thomas B. Ryerson[3], Steven S. Brown[3,8], Zachary C.J. Decker[3,4,8], Guillaume
Kirgis[3,4], Stephen Conley[9]
[1]Program in Atmospheric and Oceanic Sciences, Princeton University, Princeton, NJ, USA;
[2]NOAA Geophysical Fluid Dynamics Laboratory, Princeton, NJ, USA;
[3]NOAA Chemical Science Laboratory, Boulder, CO, USA;
[4]Cooperative Institute for Research in Environmental Sciences, University of Colorado, Boulder, CO,
USA;
[5]Department of Earth and Atmospheric Sciences, University of Houston, Houston, TX, USA;
[6]NOAA Global Monitoring Laboratory, Boulder, CO, USA
[7]C&D Technologies Inc., Philadelphia, PA, USA;
[8]Department of Chemistry, University of Colorado, Boulder, CO, USA;
[9]Scientific Aviation Inc., Boulder, CO, USA
**Corresponding author:** Meiyun Lin (meiyun.lin@noaa.gov)
**Abstract**
The detection and attribution of high background ozone ($O_3$) events in the southwestern U.S. is
challenging but relevant to the effective implementation of the lowered National Ambient Air Quality
Standard (NAAQS; 70 ppbv). Here we leverage intensive field measurements from the Fires, Asian, and
Stratospheric Transport–Las Vegas Ozone Study (*FAST*-LVOS) in May–June 2017, alongside high-
resolution simulations with two global models (GFDL-AM4 and GEOS-Chem), to study the sources of
$O_3$ during high-$O_3$ events. We show possible stratospheric influence on four out of the ten events with
daily maximum 8-hour average (MDA8) surface $O_3$ above 65 ppbv in the greater Las Vegas region. While
$O_3$ produced from regional anthropogenic emissions dominates pollution events in the Las Vegas Valley,
stratospheric intrusions can mix with regional pollution to push surface $O_3$ above 70 ppbv. GFDL-AM4
captures the key characteristics of deep stratospheric intrusions consistent with ozonesondes, lidar profiles,
and co-located measurements of $O_3$, CO, and water vapor at Angel Peak, whereas GEOS-Chem has
difficulty simulating the observed features and underestimates observed $O_3$ by ~20 ppbv at the surface.
On days when observed MDA8 $O_3$ exceeds 65 ppbv and AM4 stratospheric ozone tracer shows 20-40
ppbv enhancements, GEOS-Chem simulates ~15 ppbv lower U.S. background $O_3$ than GFDL-AM4. The
two models also differ substantially during a wildfire event, with GEOS-Chem estimating ~15 ppbv
greater $O_3$, in better agreement with lidar observations. At the surface, the two models bracket the observed
MDA8 $O_3$ values during the wildfire event. Both models capture the large-scale transport of Asian
pollution, but neither resolves some fine-scale pollution plumes, as evidenced by aerosol backscatter,
aircraft, and satellite measurements. U.S. background $O_3$ estimates from the two models differ by 5 ppbv
on average (greater in GFDL-AM4) and up to 15 ppbv episodically. Uncertainties remain in the
quantitative attribution of each event. Nevertheless, our multi-model approach tied closely to
observational analysis yields some process insights, suggesting that elevated background $O_3$ may pose
challenges to achieving a potentially lower NAAQS level (e.g., 65 ppbv) in the southwestern U.S.
**Keywords:** background ozone, stratospheric intrusions, wildfires, Asian pollution
**1 Introduction**
Surface ozone ($O_3$) typically peaks over the high-elevation southwestern U.S. (SWUS) in late spring, in
contrast to the summer maximum produced from regional anthropogenic emissions in the low-elevation
eastern U.S. (EUS). The springtime $O_3$ peak in the SWUS partly reflects the substantial influence of
background $O_3$ from natural sources (e.g., stratospheric intrusions) and intercontinental pollution (Zhang
et al., 2008; Fiore et al., 2014; Jaffe et al., 2018). These "non-controllable" $O_3$ sources can episodically
push surface daily maximum 8-hour average (MDA8) $O_3$ to exceed the NAAQS (Lin et al., 2012a; Lin et
al., 2012b; Langford et al., 2017). Identifying and quantifying the sources of springtime high-$O_3$ events in
the SWUS has been extremely challenging owing to limited measurements, complex topography, and
various $O_3$ sources (Langford et al., 2015). As the $O_3$ NAAQS becomes more stringent (lowered from 75
ppbv to 70 ppbv since 2015), quantitative understanding of background $O_3$ sources is of great importance
for screening exceptional events, i.e., "unusual or naturally occurring events that can affect air quality but
are not reasonably controllable using techniques that tribal, state or local air agencies may implement"
(U.S. Environmental Protection Agency, 2016). Here we leverage intensive measurements from the 2017
Fires, Asian, and Stratospheric Transport-Las Vegas Ozone Study (*FAST*-LVOS; Langford et al.,
manuscript in preparation), alongside high-resolution simulations with two global atmospheric chemistry
models (GFDL-AM4 and GEOS-Chem), to characterize the sources of high-$O_3$ events in the region.
Through a process-oriented analysis, we aim to understand the similarities and disparities between these
two widely-used global models in simulating $O_3$ in the SWUS.
Mounting evidence shows that a variety of sources contribute to the high surface $O_3$ found in the SWUS
during spring. For example, observational and modelling studies show that deep stratospheric intrusions
can episodically increase springtime MDA8 $O_3$ levels at high-elevation SWUS sites by 20–40 ppbv
(Langford et al., 2009; Lin et al., 2012a). Large-scale transport of Asian pollution across the North Pacific
also peaks in spring due to active mid-latitude cyclones and strong westerly winds, contributing to some
high-$O_3$ events and raising mean background $O_3$ levels over the SWUS (Jacob et al., 1999; Lin et al.,
2012b; Lin et al., 2015b; Langford et al., 2017; Lin et al., 2017). Moreover, frequent wildfires complicate
the study of $O_3$ in the SWUS (Jaffe et al., 2013; Baylon et al., 2016; Lin et al., 2017; Jaffe et al., 2018). In
the late spring and early summer, increased photochemical activity from U.S. domestic anthropogenic
emissions can prevent the unambiguous attribution of observed high-$O_3$ events in this region to
background influence.
Quantifying the contributions of different $O_3$ sources relies heavily on numerical models. Previous studies,
however, have shown large model discrepancies in the estimates of North American background $O_3$
(NAB), defined as $O_3$ that would exist in the absence of North American anthropogenic emissions. Zhang
et al. (2011) applied GEOS-Chem to quantify NAB $O_3$ during March–August of 2006–2008 and estimated
a mean of 40±7 ppbv at SWUS high-elevation sites, while Lin et al. (2012a) estimated 50±11 ppbv for the
late spring to early summer of 2010 with GFDL-AM3. Emery et al. (2012) estimated mean NAB $O_3$ to be
20–45 ppbv with GEOS-Chem and 25–50 ppbv with a regional model driven by GEOS-Chem boundary
conditions, during spring-summer. Large inter-model differences exist not only in seasonal means but also
in day-to-day variability (e.g., Fiore et al., 2014; Dolwick et al., 2015; Jaffe et al., 2018). An event-oriented
multi-model comparison, tied closely to intensive field measurements, is needed to provide process
insights into this model discrepancy.
Deploying targeted measurements and conducting robust model source attribution are crucial to
characterize and quantify the sources of elevated springtime $O_3$ in the SWUS (Langford et al., 2009;
Langford et al., 2012; Lin et al., 2012a; Lin et al., 2012b). This is particularly true for inland areas of the
SWUS, such as greater Las Vegas, where air quality monitoring sites are sparse, making it difficult to
assess the robustness of model source attribution (Langford et al., 2015; Langford et al., 2017). Using
field measurements from the Las Vegas Ozone Study (LVOS) in May–June 2013 and model simulations,
Langford et al. (2017) provided an unprecedented view of the influences of stratosphere-to-troposphere
transport (STT) and Asian pollution on the exceedances of surface $O_3$ in Clark County, Nevada. This study
suggests that $O_3$ descending from the stratosphere and sometimes mingled with Asian pollution can be
entrained into the convective boundary layer and episodically brought down to the ground in the Las
Vegas area in spring, adding 20–40 ppbv to surface $O_3$ and pushing MDA8 $O_3$ above the NAAQS.
However, uncertainties remain in previous analyses due to the use of relatively coarse-resolution
simulations and limited measurements to connect surface $O_3$ exceedances at high-elevation baseline sites
and low-elevation regulatory sites. High-resolution simulations and more extensive observations are thus
needed to further advance our understanding of springtime peak $O_3$ episodes in the region.
In May–June 2017, the NOAA Earth System Research Laboratory Chemical Sciences Division
(NOAA/ESRL CSD) carried out the *FAST*-LVOS follow up study in Clark County, NV. During this
campaign, a broad suite of near-continuous observations was collected by in situ chemistry sensors
deployed at a mountain-top site and by state-of-the-art ozone and Doppler lidars located in the Las Vegas
Valley. These daily measurements were supplemented by ozonesondes and scientific aircraft flights
during four 2- to 4-day-long intensive operating periods (IOPs) triggered by the appearance of upper-level
troughs above the U.S. West Coast. These extensive measurements, together with high-resolution
simulations from two global models (GFDL-AM4 and GEOS-Chem), provide us with a rare opportunity
to pinpoint the sources of elevated springtime $O_3$ in the SWUS. We briefly describe the *FAST*-LVOS field
campaign and model configurations in Sect. 2. Following an overall model evaluation (Sect. 3), we present
process-oriented analyses of the high-$O_3$ events from deep stratospheric intrusions, wildfires, regional
anthropogenic pollution, and the long-range transport of Asian pollution (Sect. 4). Sect. 5 summarizes
differences between the simulated total and background $O_3$ determined by the two models during *FAST*-
LVOS. Finally, in Sect. 6, the implications of the study are discussed.
**2 Measurements and Models**

## 2.1 *FAST*-LVOS measurement campaign

[Figure 1 about here]

The *FAST*-LVOS experiment was designed to further our understanding of the impacts of STT, wildfires, long-range transport from Asia, and regional pollution on air quality in the Las Vegas Valley. The field campaign was carried out between May 17 and June 30, 2017 in Clark County (NV), which includes the greater Las Vegas area (Fig. 1). The measurement campaign consisted of daily lidar and in situ measurements supplemented by aircraft and ozonesonde profiling during the four IOPs (May 23−25, May 31−June 2, June 10−14, and June 28−30). The daily measurements included chemical composition (e.g., CO and $O_3$) and meteorological parameters (e.g., air temperature and water vapor) recorded with high temporal resolution by instruments installed in a mobile laboratory (Wild et al., 2017) parked on the summit of Angel Peak (36.32°N, 115.57°W, 2682 m above sea level, a.s.l.), the site of the 2013 LVOS field campaign. This mountain-top site, located ~45 km northwest of the Las Vegas City (see Fig. 1), is far from anthropogenic emission sources and mostly receives free tropospheric air at night, but is frequently influenced during the day by air transported from the Las Vegas Valley through upslope flow in late spring and summer (Langford et al., 2015). The Tunable Optical Profiler for Aerosols and oZone (TOPAZ) 3-wavelength mobile differential absorption lidar (DIAL) system, which was previously deployed to Angel Peak during LVOS, was relocated to the North Las Vegas Airport (NLVA, Fig. 1), where it measured 8-minute averaged vertical profiles of $O_3$ and aerosol backscatter from 27.5 m to ~8 km above ground level (a.g.l.) with an effective vertical resolution (for $O_3$) ranging from ~10 m near the surface to ~150 m at 500 m a.g.l. and ~900 m at 6 km a.g.l. The aerosol backscatter profiles were retrieved at 7.5 m resolution. TOPAZ was operated daily, but not continuously, throughout the campaign. NOAA also deployed a continuously operating micro-Doppler lidar at NLVA to measure vertical velocities and relative aerosol backscatter throughout the campaign. Boundary layer heights were inferred from the micro-Doppler measurements following the method in Bonin et al. (2018).

The routine in situ and lidar measurements described above were augmented during the four IOPs by ozonesondes launched up to four times per day (30 launches total during the entire campaign) from the Clark County Department of Air Quality Joe Neal monitoring site located ~8 km north-northwest of the NLVA. Aircraft measurements were also conducted by Scientific Aviation to sample $O_3$, methane ($CH_4$), water vapor ($H_2O$), and nitrogen dioxide ($NO_2$) between NLVA and Big Bear, CA during the IOPs. Readers can refer to our previous studies (Langford et al., 2010; Alvarez II et al., 2011; Langford et al.,

2015; Langford et al., 2017; Langford et al., 2019) for detailed descriptions and configurations of the
TOPAZ and the other measurement instruments. The *FAST*-LVOS field campaign is also described in
more detail elsewhere (Langford et al., manuscript in preparation).
The *FAST*-LVOS measurements were augmented by hourly surface $O_3$ measurements from Joe Neal and
other regulatory air quality monitoring sites operated by the Clark County Department of Air Quality
(Table S1). Surface observations of $O_3$ from these and other mostly urban sites were obtained from the
U.S. Environmental Protection Agency (EPA) Air Quality System (AQS; https//www.epa.gov/aqs). We
average the AQS measurements into $0.5° \times 0.625°$ grids for a direct comparison with model results (as in
Lin et al., 2012a, b). Surface observations from rural sites and more representative of background air were
obtained from the EPA Clean Air Status and Trends Network (CASTNet; https//www.epa.gov/castnet).

## 2.2 GFDL-AM4 and GEOS-Chem

Comparisons of key model configurations are shown in Table S2. AM4 is the new generation of the
Geophysical Fluid Dynamics Laboratory chemistry-climate model contributing to the Coupled Model
Intercomparison Project, Phase 6 (CMIP6). The model employed in this study, a prototype version of
AM4.1 (Horowitz et al., 2020), differs from the AM4 configuration described in Zhao et al. (2018a, 2018b)
by including 49 vertical levels extending up to 1 Pa (~80 km) and interactive stratosphere-troposphere
chemistry and aerosols. Major physical improvements in GFDL-AM4, compared to its predecessor
GFDL-AM3 (Donner et al., 2011), include a new double-plume convection scheme with improved
representation of convective scavenging of soluble tracers, new mountain drag parametrization, and the
updated hydrostatic $FV^3$ cubed-sphere dynamical core (Zhao et al., 2016; Zhao et al., 2018a, b). For
tropospheric chemistry, GFDL-AM4 includes improved treatment of photo-oxidation of biogenic VOCs,
photolysis rates, heterogeneous chemistry, sulfate and nitrate chemistry, and deposition processes (Mao
et al., 2013a; Mao et al., 2013b; Paulot et al., 2016; Li et al., 2016; Paulot et al., 2017), as described in
more detail by Schnell et al. (2018). We implement a stratospheric $O_3$ tracer ($O_3$Strat) in GFDL-AM4 to
track $O_3$ originating from the stratosphere. The $O_3$Strat is defined relative to a dynamically varying e90
tropopause (Prather et al., 2011) and is subject to tropospheric chemical loss (in the same manner as odd
oxygen of tropospheric origin) and deposition to the surface (Lin et al., 2012a; Lin et al., 2015a). The
model is nudged to NCEP reanalysis winds using a height-dependent nudging technique (Lin et al., 2012b).
The nudging minimizes the influences of chemistry-climate feedbacks and ensures that the large-scale
meteorological conditions are similar to those observed, across the sensitivity simulations. We conduct a
suite of AM4 simulations at C192 (~50×50 km$^2$) horizontal resolution for January–June 2017: (1) a BASE
simulation with all emissions included; (2) a sensitivity simulation without anthropogenic emissions over
North America (15°–90°N, 165°–50°W; NAB); (3) a sensitivity simulation without anthropogenic
emissions over the U.S. (USB); (4) a sensitivity simulation without Asian anthropogenic emissions, and
(5) a sensitivity simulation without wildfire emissions (see Table S3). The high-resolution BASE and
sensitivity simulations for January–June 2017 are initialized from the corresponding nudged C96
(~100×100 km$^2$) simulations spanning from 2009 to 2016 (8 years). Compared to the NAB simulation,
the USB simulation includes additional contributions from Canadian and Mexican anthropogenic
emissions. The USB estimates are now generically defined as "background O$_3$" and used by the U.S. EPA.
Over the WUS, the vertical model resolution ranges from ~50–200 m near the surface to ~1–1.5 km near
the tropopause and ~2–3 km in much of the stratosphere.
Goddard Earth Observing System coupled with Chemistry (GEOS-Chem; http://geos-chem.org) is a
widely used global chemical transport model (CTM) for simulating atmospheric composition and air
quality (Bey et al., 2001; Zhang et al., 2011), driven by assimilated meteorological fields from the NASA
Global Modeling and Assimilation Office (GMAO). We conduct high-resolution simulations over North
America (10°–70°N, 140°–40°W), with 0.25° (latitude) × 0.3125° (longitude) horizontal resolution, using
a one-way nested-grid version of GEOS-Chem (v11.01) (Wang et al., 2004; Chen et al., 2009) driven by
the Goddard Earth Observing System – Forward Processing (GEOS-FP) assimilated meteorological data.
The model uses a fully coupled NO$_X$-O$_X$-hydrocarbon-aerosol-bromine chemistry mechanism in the
troposphere ("Tropchem"), whereas a simplified linearized chemistry mechanism (Linoz) is used in the
stratosphere to simulate stratospheric ozone and cross-tropopause ozone fluxes (McLinden et al., 2000).
Although GEOS-Chem can also be run with the Universal tropospheric-stratospheric Chemistry eXtension
(UCX) mechanism that simulates interactive stratosphere-troposphere chemistry and aerosols (Eastham
et al., 2014), this option was not used in the simulations presented in this study due to computational
constraints. To further save computational resources, we used a reduced vertical resolution of 47 hybrid
eta levels, by combining vertical layers above ~80 hPa from the native 72 levels of GEOS-FP. The
thickness of model vertical layers over the WUS ranges from ~15–100 m near the surface to ~1 km near
the tropopause and in the lower stratosphere. Similar GEOS-Chem simulations with simplified treatments
of stratospheric chemistry and dynamics have been previously used to estimate background O$_3$ for U.S.
EPA policy assessments (Zhang et al., 2011; Zhang et al., 2014; Fiore et al., 2014; Guo et al., 2018). Thus,

it is important to assess the ability of this model to represent high-background-$O_3$ events from stratospheric intrusions. We conduct two nested high-resolution simulations with GEOS-Chem for February−June 2017: BASE and a USB simulation with anthropogenic emissions zeroed out in the U.S. (Table S3). Initial and boundary conditions for chemical fields in the nested-grid simulations were provided by the corresponding BASE and USB GEOS-Chem global simulations at $2° \times 2.5°$ resolution for January−June 2017. Only results for April−June from the nested simulations are analyzed in this study. The three-month spin-up period (January−March) used for GEOS-Chem is relatively short compared to the multi-year GFDL-AM4 simulations, although it should be sufficient given that the lifetime of ozone in the free troposphere is approximately three weeks (e.g., Young et al., 2018).

## 2.3 Emissions

The anthropogenic emissions used in GFDL-AM4 are modified from the CMIP6 historical emission inventory (Hoesly et al., 2018). The CMIP6 emission inventory does not capture the decreasing trend in anthropogenic $NO_X$ emissions over China after 2011 as inferred from satellite-measured tropospheric $NO_2$ columns (Liu et al., 2016; Fig. S1). We thus scale CMIP6 $NO_X$ emissions over China after 2011 based on a regional emission inventory developed by Tsinghua University (personal communications with Qiang Zhang at Tsinghua University; Fig. S1). The adjusted $NO_X$ emission trend over China agrees well with the $NO_2$ trend derived from satellite retrievals. We also reduce $NO_X$ emissions over the EUS (25°−50° N, 94.5°−75° W) by 50% following Travis et al. (2016), who suggested that excessive $NO_X$ emissions may be responsible for the common model biases in simulating $O_3$ over the southeastern U.S. These emission adjustments reduce mean MDA8 $O_3$ biases in GFDL-AM4 by ~5 ppbv in spring and ~10 ppbv in summer over the EUS (Fig. S2). The model applies the latest daily-resolving global fire emission inventory from NCAR (FINN) (Wiedinmyer et al., 2011), vertically distributed over six ecosystem-dependent altitude layers from the ground surface to 6 km (Dentener et al., 2006; Lin et al., 2012b). Biogenic isoprene emissions (based on MEGAN; Guenther et al., 2006; Rasmussen et al., 2012), lightning $NO_x$ emissions, dimethyl sulfide, and sea salt emissions are tied to model meteorological fields (Donner et al., 2011; Naik et al., 2013).

For GEOS-Chem, anthropogenic emissions over the United States are scaled from the 2011 U.S. NEI to reflect the conditions in 2017 (https://www.epa.gov/air-emissions-inventories/air-pollutant-emissions-trends-data). Similar to AM4, we reduce EUS anthropogenic $NO_X$ emissions in GEOS-Chem by 50% to

improve simulated $O_3$ distributions. Anthropogenic emissions over China are based on the 2010 MIX
emission inventory (Li et al., 2017), with $NO_X$ emissions scaled after 2010 using the same trend as in
GFDL-AM4. Biogenic VOC emissions are calculated online with MEGAN (Guenther et al., 2006).
Biomass burning emissions are from the FINN inventory but implemented in the lowest model layer. The
model calculates lightning $NO_X$ emissions using a monthly climatology of satellite lightning observations
coupled to parameterized deep convection (Murray et al., 2012). The calculation of lightning $NO_X$ in this
study differs from that in Zhang et al. (2014), who used the U.S. National Lightning Detection Network
(NLDN) data to constrain model flash rates.
**3 Overall model evaluation**
**3.1 GFDL-AM4 versus GFDL-AM3**
**[Figure 2 about here]**
We first compare $O_3$ simulations in AM4 with those from its predecessor, AM3, which has been
extensively used in previous studies to estimate background $O_3$ (Lin et al., 2012a; Lin et al., 2012b; Fiore
et al., 2014; Lin et al., 2015a). Figure 2 shows the comparisons of simulated and observed March mean
$O_3$ vertical profiles and mid-tropospheric $O_3$ seasonal cycles at the Trinidad Head and Boulder ozonesonde
sites. Free tropospheric $O_3$ measured at both sites in March is representative of background conditions,
with little influence from U.S. anthropogenic emissions. Thus, we also show $O_3$ from the NAB simulations
with North American anthropogenic emissions zeroed out. As constrained by the availability of AM3
simulations from previous studies, we focus on the 2010–2014 period and compare the NAB estimates as
opposed to the USB estimates used in the rest of the paper. Compared with AM3, simulations of free
tropospheric $O_3$ are much improved in AM4. Mean $O_3$ biases are reduced by 10–25 ppbv in the middle
troposphere and 20–65 ppbv in the upper troposphere in AM4, reflecting mostly an improved simulation
of background $O_3$ (Fig. 2a). These improvements are mainly credited to the changes in
dynamics/convection schemes in AM4 (Zhao et al., 2018a), according to our sensitivity simulations (not
shown). The difference in emissions inventories contribute to some of the $O_3$ differences but is not the
major cause because the largest differences between the two models in simulated free tropospheric $O_3$
occur during the cold months (November-April) when photochemistry is weak (Fig.2b).
**3.2 GFDL-AM4 versus GEOS-Chem**
**[Figure 3 about here]**
Next, we examine how GFDL-AM4 compares with GEOS-Chem in simulating the mean distribution and
the day-to-day variability of total and USB $O_3$ in the free troposphere (Fig. 3) and at the surface (Fig. 4
and Fig. S3) during *FAST*-LVOS. Comparisons with ozonesondes at Joe Neal show that the total $O_3$
concentrations below 700 hPa simulated by the two models often bracket the observed values (Fig. 3a).
Between 700–300 hPa, GFDL-AM4 better captures the observed mean and day-to-day variability of $O_3$,
as evaluated with the standard deviation. Further comparison with lidar measurements averaged over 3–6
km altitude above Las Vegas shows that total and USB $O_3$ in GFDL-AM4 exhibits larger day-to-day
variability than in GEOS-Chem ($\sigma$ = 8.1 ppbv in observations, 8.1 ppbv in AM4, and 6.7 ppbv in GEOS-
Chem; Fig. 3c). For mean $O_3$ levels in the free troposphere, AM4 estimates a 7 ppbv contribution from
U.S. anthropogenic emissions (total minus USB), while GEOS-Chem suggests only 3.5 ppbv. The largest
discrepancies between the two models occurred on June 11–13 (the blue shaded period in Fig. 3c), which
we later attribute to a stratospheric intrusion event (Sect. 4). During this period, AM4 simulates elevated
$O_3$ (70–75 ppbv) broadly consistent with the lidar and sonde measurements, while GEOS-Chem
considerably underestimates the observations by 20 ppbv. Consistent with total $O_3$, USB $O_3$ in GFDL-
AM4 is much higher than GEOS-Chem during this event.
**[Figure 4 about here]**
Figure 4 shows the times series of observed and simulated surface MDA8 $O_3$ at four high-elevation sites
and one low-elevation site in the region during the study period. Statistics comparing the results at all sites
are shown in Supplementary Table S1. The two models show large differences in simulated total and USB
$O_3$ on days when the $O_3$Strat tracer in AM4 indicates stratospheric influence (highlighted in blue shading).
AM4 $O_3$Strat indicates frequent STT events during April–June, with observed MDA8 $O_3$ exceeding or
approaching the current NAAQS of 70 ppbv. Compared with observations, GFDL-AM4 captures the
spikes of MDA8 $O_3$ and elevated USB $O_3$ during these STT events (e.g., April 23, May 13, and June 11).
On these days, GEOS-Chem underestimates observed $O_3$ by 10–25 ppbv and simulates much lower USB
$O_3$ levels than GFDL-AM4. For some days, GFDL-AM4 overestimates total MDA8 $O_3$ due to excessive
STT influence (e.g., May 7 at Spring Mountain Youth Camp). The two models also differ substantially in
total and USB $O_3$ (14–18 ppbv) on June 22 (yellow shading), with GEOS-Chem overestimating
observations at high-elevation sites while GFDL-AM4 underestimates observations at both high- and low-
elevation sites. We provide more in-depth analysis of these events in Sect. 4 and identify the possible
causes of the model biases.

## 4 Process-oriented analysis of high-ozone events during *FAST*-LVOS

**[Table 1 about here]**

We identify ten events with observed MDA8 $O_3$ exceeding 65 ppbv at multiple sites in the greater Las Vegas area during April–June 2017. Table 1 provides an overview of the events, the dominant source for each event, the surface sites impacted, and the associated analysis figures presented in this article. Observations and model simulations of MDA8 $O_3$ for each event are also included in Table 1 for Angel Peak and in Supplementary Table S4 and Fig.S4 for all Clark County surface sites. The attribution is based on a combination of observational and modeling analyses. First, we examine the $O_3$/CO/$H_2O$ relationships and collocated meteorological measurements from the NOAA/ESRL mobile lab deployed at Angel Peak to provide a first guess on the possible sources of the observed high-$O_3$ events (Sect. 4.1). Then, we analyze large-scale meteorological fields (e.g., potential vorticity), satellite images (e.g., AIRS CO), and lidar and ozonesonde observations to examine if the transport patterns, the high-$O_3$ layers, and related tracers are consistent with the key characteristics of a particular source (Sect. 4.2–4.5). Available aerosol backscatter measurements and multi-tracer aircraft profiles are also used to support the attribution (Sect. 4.3 and 4.6). Finally, for each event we examine the spatiotemporal correlations of model simulations of total $O_3$, background $O_3$, and its components (e.g., stratospheric ozone tracer), both in the free troposphere and at the surface. For a source to be classified as the dominant driver of an event, $O_3$ from that source must be elevated sufficiently from its mean baseline value.

### 4.1 Observed $O_3$/CO/$H_2O$ relationships

**[Figures 5-6 about here]**

Relationships between concurrently measured $O_3$ and CO are useful to identify the possible origins of elevated surface $O_3$ (Parrish et al., 1998; Herman et al., 1999; Langford et al., 2015). During *FAST*-LVOS, in-situ 1-min measurements at Angel Peak show differences in $\Delta O_3/\Delta CO$ and water vapor content between air plumes during a variety of events (Figs. 5, 6, and S5). Notably, on June 11, $O_3$ was negatively correlated with CO ($\Delta O_3/\Delta CO$ = -3.79). This anti-correlation is distinctly different from the $O_3$/CO relationships during other periods (e.g., $\Delta O_3/\Delta CO$ = 0.68–0.70 on June 16 or $\Delta O_3/\Delta CO$ = 1.08 on June 2). The negative correlation (high $O_3$ together with low CO) serves as strong evidence of a stratospheric origin of the air masses on June 11, since $O_3$ is much more abundant in the stratosphere than in the troposphere whereas CO is mostly concentrated within the troposphere where it is directly emitted or chemically formed

(Langford et al., 2015). On the contrary, simultaneously elevated $O_3$ and CO suggest influences by
wildfires (e.g., June 22) or anthropogenic (e.g., June 16) pollution (Figs. 6b–d and S4). In particular,
exceptionally high CO levels (~100–440 ppbv) on June 22 (Fig. 6e) suggest influences from wildfires.
Ozone enhancements were measured by the TOPAZ ozone lidar on June 22 (Sect. 4.3), although the
correlation between CO and $O_3$ at Angel Peak is not strong. The net production of $O_3$ by wildfires is highly
variable, with many contradictory observations reported in the literature (Jaffe and Wigder, 2012). The
amount of $O_3$ within a given smoke plume varies with distance from the fire and depends on the plume
injection height, smoke density, and cloud cover (Faloona et al., 2020).
We gain further insights by examining water vapor concurrently measured at Angel Peak. Air masses from
the lower stratosphere are generally dry, whereas wildfire/urban plumes from the boundary layer are
relatively moist (Langford et al., 2015). Thus, the dry conditions of the air masses on June 11 support our
conclusion that the plume was transported downward from the upper troposphere and lower stratosphere
(Fig. 6a). These conditions are in contrast to those of the urban/wildfire plumes transported from the Las
Vegas Valley (Fig. 6c–6d). Additionally, we separate the anthropogenic plumes on June 16 into daytime
and nighttime conditions because of a diurnal variation of air conditions (relatively dry at night versus wet
during daytime; Figs. 6c–d). This analysis further demonstrates that the anthropogenic pollution plume
during nighttime is wetter than the stratospheric air on June 11. On June 14 (Fig. 6b), measured $O_3$ was
positively correlated with CO, indicating regional/local pollution influence, but the lower levels of water
vapor than those in regional pollution and wildfire plumes suggest that the stratospheric air which reached
Angel Peak earlier may have been mixed with local pollution. On June 28 (Fig. 6f), $O_3$ was positively
correlated with CO and the air masses were relatively dry, indicating that the plume was likely from aged
pollution transported from Asia or Southern California as opposed to from fresh pollution from the Las
Vegas Valley. Identifying the primary source of the high-$O_3$ events solely based on observations is
challenging; additional insights from models are thus needed as we demonstrate below.
**4.2 Characteristics of stratospheric intrusion during June 11−14**
**[Figures 7-8 about here]**
Analysis of the 250 hPa potential vorticity and the AM4 model stratospheric $O_3$ tracer shows significant
stratospheric influence on surface $O_3$ in the SWUS on April 22−23 (Fig. S6), May 13−14 (Fig. S6), and
June 11−14 (Figs. 7−8). During these events, surface MDA8 $O_3$Strat in AM4 was 20-40 ppbv higher than
the mean baseline level (15–20 ppbv; see dashed purple lines Fig. 4). Below, we focus on the June 11–14
event, which was the subject of a 4-day *FAST*-LVOS IOP with 60 hours of continuous $O_3$ lidar profiling
and 13 ozonesonde launches, in addition to continuous in situ measurements at Angel Peak.

*Deep stratospheric intrusion on June 11–13*

Synoptic-scale patterns of potential vorticity (PV) indicate a strong upper-level trough over the northwest
U.S. on June 12 (PV = 4–5 PVU in Fig. 7a). The PV pattern displays a "hook-shaped" streamer of air
extending from the northern U.S. to the Intermountain West, a typical feature for a STT event (Lin et al.,
2012a; Akritidis et al., 2018). This upper-level trough penetrated southeastwardly towards the SWUS,
facilitating the descent of stratospheric air masses into the lower troposphere. Ozonesondes launched at
Joe Neal on June 12 recorded elevated $O_3$ levels of 150–270 ppbv at 5–8 km altitude (color-coded circles
in Fig. 7b). Consistent with the ozonesonde measurements, GFDL-AM4 shows that $O_3$-rich stratospheric
air masses descended isentropically towards the study region, with simulated $O_3$ reaching 90 ppbv at ~2
km altitude. For comparison, GEOS-Chem simulates a much weaker and shallower intrusion (Fig. 7b),
despite a similar synoptic-scale pattern of potential vorticity at 250 hPa and comparable ozone levels in
the UTLS (Fig. S7), suggesting possibly greater numerical diffusion in GEOS-Chem diluting the
stratospheric intrusion. There are also some notable differences in the isentropic surfaces (e.g., at 322 K)
between the two models, possibly resulting from a difference in the two meteorological reanalysis data
(NCEP in AM4 and MERRA in GEOS-Chem).
TOPAZ lidar measurements at NLVA vividly characterize the strength and vertical depth of intruding $O_3$
tongues evolving with time (Fig. 8a). A tongue of high $O_3$ exceeding 100 ppbv descended to as low as 2–3
km altitude on June 12. GFDL-AM4 captures both the timing and structure of the observed high-$O_3$ layer
and attributes it to a stratospheric origin as supported by the $O_3$Strat tracer. In contrast, GEOS-Chem
substantially underestimates the depth and magnitude of the observed high-$O_3$ layers in the free
troposphere. Zhang et al. (2014) also showed that GEOS-Chem captures the timing of stratospheric
intrusions but underestimates their magnitude by a factor of 3.
**[Figure 9 about here]**
Surface observations show that high MDA8 $O_3$ exceeding 60 ppbv first emerged on June 11 over Southern
Nevada (Fig. 9), consistent with the arrival of stratospheric air masses as inferred from the negative
correlation between $O_3$ and CO measured at Angel Peak (Fig. 6a). Over the next few days, the areas with
observed MDA8 $O_3$ approaching 70 ppbv gradually shifted southward from Nevada and Colorado to
Arizona and New Mexico. By June 13, observed surface MDA8 $O_3$ exceeded 70 ppbv over a large
proportion of the SWUS, including Arizona and New Mexico. GFDL-AM4 captures well the observed
day-to-day variability of high-$O_3$ spots over the WUS, although the model overall has high biases. Over
the areas where observed MDA8 $O_3$ levels are 60–75 ppbv, GFDL-AM4 estimates 50–65 ppbv USB $O_3$
with simulated $O_3$Strat 20–40 ppbv higher than its mean baseline level in June. GEOS-Chem has difficulty
simulating the observed high-$O_3$ areas during this event and simulated USB is 15 ppbv lower than AM4
(Fig. 9). These results are consistent with the fact that GEOS-Chem does not capture the structure and
magnitude of deep stratospheric intrusions during the period (Figs. 3, 7, and 8).
*Mixing of stratospheric ozone with regional pollution on June 14*
Stratospheric air masses that penetrate deep into the troposphere can mix with regional anthropogenic
pollution and gradually lose their typical stratospheric characteristics (cold and dry air containing low
levels of CO), challenging diagnosis of stratospheric impacts based directly on observations (Cooper et
al., 2004; Lin et al., 2012b; Trickl et al., 2016). On June 14, $O_3$ measured at Angel Peak is positively
correlated with CO ($\Delta O_3/\Delta CO = 0.75$; Fig. 6b), similar to conditions of anthropogenic pollution on June
16 (Fig. 6c–d). TOPAZ lidar shows elevated $O_3$ of 70–80 ppbv concentrated within the boundary layer
below 3 km altitude (Fig. 8b). These observational data do not provide compelling evidence for
stratospheric influence. However, GFDL-AM4 simulates elevated $O_3$Strat coinciding with the observed
and modeled total $O_3$ enhancements within the PBL, indicating that $O_3$ from the deep stratospheric
intrusion on the previous day may have been mixed with regional anthropogenic pollution to elevate $O_3$
in the PBL. At the surface (the bottom panel in Fig. 9), AM4 simulates high USB $O_3$ and elevated $O_3$Strat
(20–40 ppbv above its mean baseline) over Arizona and New Mexico where MDA8 $O_3$ greater than 70
ppbv was observed. The fact that GEOS-Chem is unable to simulate the ozone enhancements in lidar
measurements and at the surface further supports the possible stratospheric influence. This case study
demonstrates the value of integrating observational and modeling analysis for the attribution of high-$O_3$
events over a region with complex $O_3$ sources.
The extent to which stratospheric intrusions contribute to surface $O_3$ at low-elevation sites over the WUS
is poorly characterized in previous studies. Notably, surface $O_3$ at three low-elevation (~700–800 m a.s.l.)
air quality monitoring sites in Clark County exceeded the current NAAQS level of 70 ppbv on June 14:
74 ppbv at Joe Neal, 73 ppbv at North Las Vegas Airport, and 71 ppbv at Walter Johnson. The number of
monitoring sites with $O_3$ exceedances would have increased to eleven in Clark County if the NAAQS had
been lowered to 65 ppbv. While $O_3$ produced from regional anthropogenic emissions still dominates
pollution in the Las Vegas Valley (Fig. S4), our analysis shows that stratospheric intrusions can mix with
regional pollution to push surface $O_3$ above the NAAQS.

**4.3 Wildfires on June 22**

**[Figure 10 about here: Aerosol backscatter]**

[Figure 11 about here]

Significant enhancements in aerosol backscatter were observed at 3−6 km altitude above NLVA on June
21−22, indicating the presence of wildfire smoke (Fig. 10a). Under the influence of the wildfire plume,
mobile lab measurements at Angel Peak (~3 km altitude) detected elevated CO as high as 440 ppbv in
warm, moist air masses (Fig. 6e). The lidar measurements at NLVA on June 22 showed broad $O_3$
enhancements (80−100 ppb) from the surface to 4 km altitude (Fig. 11a). After 12:00 PDT (19:00 UTC),
a deep PBL (3–4 km) developed and $O_3$ within the PBL was substantially enhanced (> 80 ppbv), likely
due to strong $O_3$ production through reactions between abundant VOCs in the wildfire plumes and $NO_X$
in urban environments (Singh et al., 2012; Gong et al., 2017). Surface MDA8 $O_3$ exceeded 70 ppbv at
multiple sites in the Las Vegas Valley during the event (Table 1). Unfortunately, the synoptic conditions
did not trigger an IOP, so there were no aircraft or ozonesonde measurements during this event.
GFDL-AM4 has difficulty simulating the $O_3$-rich plumes above Clark County on June 22 (Fig.11a).
GEOS-Chem captures the observed high-$O_3$ layers within the PBL, but overestimates $O_3$ above 4 km
altitude (Fig.11a). GEOS-Chem overestimates of free tropospheric ozone seem to be common for the non-
STT events during late spring  through summer (Figs.3b; Fig.8b,  Fig.11b, and comparisons with lidar
data for May 24 and June 16 shown in Sect. 4.4-4.6), likely due to excessive $O_3$ produced from lightning
$NO_x$ over the southern U.S. (Zhang et al., 2011; Zhang et al., 2014).  At the surface, total MDA8 $O_3$
concentrations simulated by the two models bracket the observed values at sites in the Las Vegas area (see
yellow shading in Fig. 4) and across the Intermountain West (Fig. 12a). AM4 does not simulate elevated
$O_3$ during this event, while GEOS-Chem simulates elevated total and USB $O_3$ levels across the entire
Southwest region. GEOS-Chem simulations during this wildfire event agree better with the observed
MDA8 $O_3$ enhancements (> 70 ppbv) at Joe Neal (Fig. 4). At the high-elevation sites Angel Peak and
Spring Mountain Youth Camp, however, GEOS-Chem overestimates the observed MDA8 $O_3$ by 10–15
ppbv. Overall, GEOS-Chem seems to be more consistent with observations than GFDL-AM4 during this
wildfire event. However, we cannot rule out the possibility that the better agreement between observations
and GEOS-Chem simulations during this event may reflect excessive $O_3$ from lightning $NO_X$ in the model
(Zhang et al., 2014).
Meteorological conditions (e.g., temperature and wind fields) on June 22 in the reanalysis data used by
GFDL-AM4 and GEOS-Chem are similar over the WUS (not shown). The two models use the same
wildfire emissions (FINN) but with different vertical distributions. Fire emissions are distributed between
the surface and 6 km altitude in GFDL-AM4 but are placed at the surface level in GEOS-Chem. We
conduct several sensitivity simulations with GFDL-AM4 to investigate the causes of the model biases.
Placing all fire emissions at the surface in GFDL-AM4 results in ±5 ppbv differences in modeled MDA8
$O_3$ on June 22 (Fig. S8). Observations suggested that 40% of $NO_X$ can be converted rapidly to PAN and
20% to $HNO_3$ in fresh boreal fire plumes over North America (Alvarado et al., 2010). Both models
currently treat 100% of wildfire $NO_X$ emissions as NO. We conduct an additional AM4 sensitivity
simulation, in which 40% of the wildfire $NO_X$ emissions are released as PAN and 20% as $HNO_3$. This
treatment results in ±2 ppbv differences in simulated monthly mean MDA8 $O_3$ during an active wildfire
season (August 2012; Fig. S9). Overall, these changes do not substantially improve simulated $O_3$ on June
22. Future efforts are needed to investigate the ability of current models to simulate $O_3$ formations in fire
plumes (Jaffe et al., 2018).
**4.4 Regional and local anthropogenic pollution events**
[Figure 12 about here]
Regional and local anthropogenic emissions were important sources of elevated $O_3$ in Clark County during
*FAST*-LVOS, contributing to three out of ten observed high-$O_3$ events above 65 ppbv during April–June
2017 (Table 1). Below, we focus on the June 16 event when severe $O_3$ pollution with MDA8 $O_3$ exceeding
70 ppbv occurred over California, Arizona, parts of Nevada, and New Mexico. Analysis for the June 2
and June 29–30 pollution events are shown in the supplemental material (Figs. S5, S10, and S11). The
TOPAZ lidar measurements on June 16 show elevated $O_3$ of 55–90 ppbv in the 4-km-deep PBL (Fig. 11b).
However, this event did not trigger an IOP, so ozonesonde and aircraft measurements are unavailable.
Both GFDL-AM4 and GEOS-Chem capture the buildup of $O_3$ pollution in the PBL on June 16 (Fig. 11b).
Both models show boundary layer enhancements of total $O_3$ but not of USB $O_3$ (Fig. 11b), indicating that
regional or local anthropogenic emissions are the primary source of observed $O_3$ enhancements. Similar
to June 16, GEOS-Chem clearly shows enhancements in total $O_3$ in the PBL but not in USB $O_3$ on June 2
and June 29−30 (Fig.S10). The model attribution to U.S. anthropogenic emissions is consistent with the
positive correlation between $O_3$ and CO measured at Angel Peak on June 16 (Fig. 6c−6d), June 2, and
June 29−30 (Fig.S5). It is noteworthy that, with its higher horizontal resolution, GEOS-Chem better
resolves the structure of the $O_3$ plumes as observed by TOPAZ lidar for all the three pollution events. At
the surface, both models capture the large-scale MDA8 $O_3$ enhancements across the SWUS on June 16
(Fig. 12b). The surface $O_3$ enhancements on June 2 and June 29−30 are relatively localized in Southern
California and the Las Vegas area (Fig. S11), and both models have difficulty simulating the observed
peak MDA8 values (Fig. 4).
**4.5 Long-range transport of Asian pollution on May 20−24**
**[Figures 13-15 about here]**
During May 20−24, long-range transport of Asian pollution toward the WUS was observed via large-scale
CO column observations with Atmospheric Infrared Sounder (AIRS) on NASA's Aqua satellite (Fig. 13a).
These Asian plumes traveled eastward across the Pacific for several days, reaching the west coast of the
U.S. on May 23 during the first *FAST*-LVOS IOP (May 23−25). The lidar measurements at NLVA on
May 24 clearly showed high-$O_3$ plumes (> 70 ppbv) concentrated within the layers of 1−4 km and 6−8
km altitude above the Las Vegas Valley throughout the day (Fig. 14a). Both GFDL-AM4 and GEOS-
Chem capture the observed $O_3$-rich plumes at surface−4 km and 6−8 km altitude above Clark County
during this event. Elevated $O_3$ at 6−8 km altitude reflects the long-range transport from Asia, as supported
by concurrent enhancements in total and USB $O_3$ in both models and by the large difference in $O_3$ between
the AM4 BASE simulation and the sensitivity simulation with Asian anthropogenic emissions zeroed out.
Elevated $O_3$ at 1−4 km altitude appears to be influenced by a residual pollution layer from the previous
day; this plume was later mixed into the growing PBL (up to 4 km altitude), elevating MDA8 $O_3$ in surface
air on May 24.  Further supporting the impact from regional or local pollution below 4 km altitude, both
models simulate much larger enhancements in total $O_3$ (70−90 ppbv) than in USB $O_3$ (~50 ppbv).
On May 24, MDA8 $O_3$ approached or exceeded the 70-ppbv NAAQS at multiple sites in California, Idaho,
Wyoming, and Nevada (Fig. 15a), likely reflecting the combined influence of regional pollution and long-
range transport of Asian pollution. MDA8 $O_3$ at four surface sites in Clark County was above 65 ppbv.
More exceedances would have occurred if the level for the NAAQS were lowered to 65 ppbv. In parts of
Idaho, Wyoming, and California where observed MDA8 $O_3$ was higher than 60 ppbv, the contribution of
Asian anthropogenic emissions as estimated by GFDL-AM4 was 8–15 ppbv (Fig. 15a), much higher than
the springtime average contribution of ~5 ppbv estimated by previous studies (e.g., Lin et al., 2012b),
supporting the episodic influence from Asian pollution during this event. At several high-elevation sites
in California such as Arden Peak (72 ppbv) and Yosemite National Park (70 ppbv), where observed MDA8
$O_3$ exceeds the NAAQS level, the contribution of Asian pollution is approximately 9 ppbv. Ozone
produced from regional and local anthropogenic emissions dominates the observed MDA8 $O_3$ above 70
ppbv in the Central Valley of California.
**4.6 An unattributed event: June 28**
The lidar measurements from June 28 show a fine-scale structure with a narrow $O_3$ layer exceeding 100
ppbv at 3–4 km altitude during 08:00–14:00 PDT (15:00–21:00 UTC shown in Fig. 14b). An ozonesonde
launched at 12:00 PDT also detected a high-$O_3$ layer (~115 ppbv) between 3.5 and 4 km altitude (not
shown). This high-$O_3$ filament appears to descend and mix into the PBL after 14:00 PDT (21:00 UTC),
contributing to elevated $O_3$ within the PBL in the afternoon. Both models are unable to represent this fine-
scale transport event, possibly due to diffusive mixing of the narrow layer (Fig. 14b). We, therefore, focus
on available airborne and in situ measurements to investigate the origin of this fine-scale $O_3$ filament.
Our examinations of large-scale satellite CO column measurements reveal a migration during June 23–27
of high-CO plumes from Asia that arrived at the west coast of the U.S. on June 27 (Fig. 13b). GFDL-AM4
estimates 5–6 ppbv contributions from Asian pollution over the WUS on June 28 (Figs. 15b), which do
not represent a significant enhancement above the mean Asian contribution. Aircraft measurements above
the Las Vegas Valley in the late morning showed collocated enhancements in $CH_4$ and $O_3$ coincident with
low free-tropospheric water vapor values at 3–4 km altitude (Fig. 10b). In-situ measurements at Angel
Peak show concurrent increases in CO and $O_3$ coincident with relatively dry conditions that are consistent
with transported Asian pollution, but these increases did not appear until several hours after the fine-
scale filament was entrained by the mixed layer (Fig. 6f). These observations indicate that the $O_3$-rich
plume appears to be unrelated to stratospheric intrusions. Aerosol backscatter measurements at NLVA
show only a slight enhancement in backscatter within the elevated $O_3$ layer on June 28, in contrast to the
thick smoke observed on June 22 when the Las Vegas Valley was influenced by fresh wildfires (Fig. 10).
HYSPLIT and FLEXPART analyses presented in Langford et al. (in preparation) suggest a possible
connection to the Schaeffer Fire (https://en.wikipedia.org/wiki/Schaeffer_Fire) in the Sequoia National
Forest in California. Another possible source is the fine-scale lofting of pollution from Southern California
followed by transport into the free troposphere over Las Vegas (Langford et al., 2010). This event further
demonstrates the complexity of $O_3$ sources in the SWUS. We recommend measurements of atmospheric
compounds like acetonitrile ($CH_3CN$, abundant in fire plumes) and methyl chloride ($CH_3Cl$, abundant in
Asian pollution) (Holzinger et al., 1999; Barletta et al., 2009) via aircraft and in situ platforms in future
field campaigns in the region to help identify the sources of such high-$O_3$ filaments.

## 5 Comparison of background ozone simulated with GFDL-AM4 and GEOS-Chem

**[Figure 16 about here]**

Here, we summarize the differences in total and background $O_3$ between the two models over the WUS.
GFDL-AM4 and GEOS-Chem differ in their spatial distributions and magnitudes of April–June mean
USB $O_3$ at the surface and in the free troposphere over the U.S. (Fig. 16 and Fig. S12). USB $O_3$ in GFDL-
AM4 peaks over the high-elevation Intermountain West at the surface (45−55 ppbv; Fig. 16a) and over
the northern U.S. in the free troposphere (3−6 km altitude; 50−65 ppbv; Fig. 16b), due to stronger STT
influence. In comparison, GEOS-Chem simulates higher USB $O_3$ levels in southwestern states (e.g.,
Texas), both at the surface (45−50 ppbv) and at 3−6 km altitude (55−65 ppbv), likely due to excessive
lightning $NO_X$ during early summer (Zhang et al., 2011; Zhang et al., 2014; Fiore et al., 2014). The
different north-south gradient in simulated USB between the two models (Fig. 16b and Fig. S12) likely
reflect that GFDL-AM4 simulates stronger STT influences over the northwestern U.S. while GEOS-Chem
produces greater $O_3$ from lightning $NO_X$ emissions in the free troposphere over the southern U.S. Despite
a quantitative disparity, both models simulate higher USB $O_3$ levels over the WUS (45−55 ppbv in GFDL-
AM4 and 35−45 ppbv in GEOS-Chem) than over the EUS at the surface (Fig. 16a). Our USB $O_3$ estimates
with GEOS-Chem are generally consistent with the estimates in previous studies using GEOS-Chem or
regional models driven by GEOS-Chem boundary conditions (Zhang et al., 2011; Emery et al., 2012;
Dolwick et al., 2015; Guo et al., 2018). In contrast to NAB $O_3$ estimates in earlier studies by zeroing out
North American anthropogenic emissions (Zhang et al., 2011; Lin et al., 2012a; Fiore et al., 2014; Zhang

et al., 2014), USB $O_3$ estimates in our study include the additional contribution from Canadian and Mexican emissions. USB $O_3$ at Clark County sites is ~4 ppbv greater than NAB $O_3$ in GFDL-AM4 (Table S5). We also find that NAB $O_3$ estimated with the new GFDL-AM4 model is ~5 ppbv lower than the NAB estimates by its predecessor GFDL-AM3 (Lin et al., 2012a) for the WUS during March−April (Fig. S13), consistent with an improved simulation of free tropospheric ozone in AM4 during spring (Fig. 2). During early summer, the NAB $O_3$ levels estimated by AM3 and AM4 are similar (Fig. S13).

**[Figure 17 about here]**

We further compare simulated surface MDA8 $O_3$ against observations at 12 high-elevation sites (> 1500 m altitude; including 11 CASTNet sites and Angel Peak; see Table S1 and black circles in Fig. 1) in the WUS (Fig. 17). The observed high-MDA8-$O_3$ events above 65 ppbv at these high-elevation sites are generally associated with enhanced background $O_3$ in both models (USB $O_3$ = 50−60 ppbv in GFDL-AM4 and 45−55 ppbv in GEOS-Chem; Fig.17a). Stratospheric intrusions are an important source of the observed events above 65 ppbv (Fig. S14), as indicated by GFDL-AM4, which better captures these high-$O_3$ events influenced by elevated background $O_3$ contributions, whereas GEOS-Chem underestimates these extreme events (comparing points in the top-right box in Fig. 17a). Although AM4 is capable of simulating most of the highest observed springtime MDA8 $O_3$ events (>65 ppbv) over the WUS, we note that AM4 tends to overestimate stratospheric influence on days when observed MDA8 $O_3$ is on the range of 50−65 ppbv. For mean MDA8 $O_3$ at these sites, GFDL-AM4 is biased high by 3 ppbv while GEOS-Chem is biased low by 5 ppbv. Mean USB $O_3$ simulated with GFDL-AM4 is 51.4±7.8 ppbv at WUS sites, higher than that in GEOS-Chem (45.7±5.7 ppbv; Fig. 17b). Probability distributions show that GFDL-AM4 simulates a wider range of total and USB $O_3$ than GEOS-Chem, reflecting relative skill in capturing the day-to-day variability of $O_3$. In addition to background $O_3$ discussed in the present study, recent studies also found that ozone dry deposition coupled to vegetation can substantially influence model simulations of surface $O_3$ means and extremes (Lin et al., 2019; Lin et al. 2020).

Tables S5 and S6 report year-to-year variability in the percentage of site-days with springtime MDA8 $O_3$ above 70 ppbv (or 65 ppbv) and simulated USB levels during 2010−2017. The percentage of site-days with MDA8 $O_3$ above 70 ppbv during April−June 2017 is 0.9% from observations at CASTNet sites, 2.0% from GFDL-AM4, and 0.1% from GEOS-Chem. GFDL-AM4 captures some aspects of the observed year-to-year variability despite mean-state biases. For example, the observed percentage of site-days with MDA8 $O_3$ above 70 ppbv at CASTNet sites is highest (9.4%) in April−June 2012, compared to 3.1±3.2%

for the 2010–2017 average. The corresponding statistics from GFDL-AM4 are 7.7% for 2012 and 4.0±2.9%
for the 2010–2017 average. The May–June mean USB MDA8 $O_3$ in GFDL-AM4 at Clark County sites
are 50.9 ppbv in 2017, 55.3 ppbv in 2012, and 52.3±2.0 ppbv for the 2010–2017 average. Supporting the
conclusions of Lin et al. (2015a), these results indicate that background $O_3$, particularly the stratospheric
influence, is an important source of the observed year-to-year variability in high-$O_3$ events over the WUS
during spring.
**6 Discussion and Conclusions**
Through a process-oriented analysis of intensive measurements from the 2017 *FAST*-LVOS field
campaign and high-resolution simulations with two global models (GFDL-AM4 and GEOS-Chem), we
study the sources of observed MDA8 $O_3$ above 65 ppbv in the SWUS. Attribution of each event to a
specific source is sometimes challenging, despite an integrated analysis of multi-tracer, multi-platform
observations and model simulations. We identify the high-$O_3$ events associated with stratospheric
intrusions (April 22–23, May 13–14, and June 11–13), mixing of local pollution and transported
stratospheric $O_3$ (June 14), regional or local anthropogenic pollution (June 2, June 16, and June 29–30),
wildfires (June 22), and mixing of Asian pollution with regional pollution (May 24). We also discuss an
event (June 28) likely resulting from the fine-scale transport of fire plumes or pollution from Southern
California, although a solid attribution for this event is challenging based on available data.
During the June 11–13 deep stratospheric intrusion event, the NOAA mobile lab measurements at Angel
Peak show a sharp increase in $O_3$ coinciding with a decrease in CO and water vapor, a marker for air of
stratospheric origin. These characteristics are in contrast to the concurrent increases in $O_3$ and CO in humid,
warm urban plumes and wildfires plumes transported from the Las Vegas Valley. The observed
$O_3$/CO/$H_2O$ relationships can provide a useful first indication of high-$O_3$ events influenced directly by a
deep intrusion. However, once transported stratospheric $O_3$ is mixed into regional pollution, model
diagnostic tracers are needed to quantify the stratospheric impact. For instance, on June 14, observations
at Angel Peak show positive $O_3$/CO correlations while $O_3$Strat in GFDL-AM4 shows 20–30 ppbv
enhancements above its mean level at Angel Peak and at surface sites across the SWUS where the
observed and simulated total MDA8 $O_3$ concentrations were above 70 ppbv. These quantitative model
attributions are only as good as the precision and capability of the models.
GFDL-AM4 and GEOS-Chem differ significantly in simulating stratosphere-to-troposphere transport
events, affecting their ability to simulate USB mean levels and extreme events. During the June 11−14
STT event, GFDL-AM4 captures the key characteristics of deep stratospheric intrusions, consistent with
lidar profiles and ozonesondes, whereas GEOS-Chem with simplified stratospheric chemistry and
dynamics has difficulty simulating the observed features. At the surface, on days when observed MDA8
$O_3$ exceeds 65 ppbv and AM4 $O_3$Strat is 20–40 ppbv above its mean baseline level, AM4 simulates 15−20
ppbv greater USB $O_3$ than GEOS-Chem (Figs. 4 and 9). During these STT events, total MDA8 $O_3$
abundances simulated by the two models often bracket the observed values, as noted previously by Fiore
et al. (2014). The *FAST*-LVOS analysis, combined with our earlier multi-year studies (Lin et al. 2012a;
Lin et al., 2015a), indicate that GFDL AM3/AM4 with nudged meteorology captures the timing and
locations of the observed $O_3$ enhancements in surface air and aloft during STT events, and is thus useful
for screening of exceptional events due to STT. AM3/AM4 typically spreads the STT enhancement across
a wider range of sites over the Southwest rather than capturing the observed localized feature, causing
high biases of total MDA8 $O_3$ during some STT events (Lin et al., 2012a). Thus, we propose targeted
analysis of the observed high-$O_3$ events, rather than the modeled events, and recommend bias correction
to simulated USB $O_3$ in AM4, such as the approach used by Lin et al. (2012a). For the future application
of GEOS-Chem for USB estimates, we recommend the version with the Universal tropospheric-
stratospheric Chemistry eXtension (UCX) mechanism (Eastham et al., 2014) and process-oriented
evaluation using daily ozonesondes and lidar profiles.
The two models also differ substantially in total and background $O_3$ simulations during the June 22
wildfire event. GEOS-Chem captures the broad $O_3$ enhancement in lidar observations, but overestimates
surface MDA8 $O_3$ at some sites during this event. It remains unclear whether the higher USB $O_3$ simulated
by GEOS-Chem during this event is from greater $O_3$ produced from wildfire emissions or excessive
lightning $NO_x$ emissions in the model. Although GFDL-AM3 captures the observed interannual variability
in $O_3$ enhancements from large-scale wildfires over the WUS (Lin et al., 2017), GFDL-AM4 has difficulty
simulating the observed $O_3$ enhancements during the relatively small-scale wildfire event on June 22.
Sensitivity simulations with fire emissions constrained at the surface or with part of fire $NO_x$ emissions
emitted as PAN and $HNO_3$ do not substantially improve simulated $O_3$ on June 22. Wildfires typically
occur under hot, dry conditions, which also enable the buildup of $O_3$ produced from regional
anthropogenic emissions, complicating an unambiguous attribution of the high-$O_3$ events solely based on
observations. Screening of exceptional events due to wildfire emissions remains a serious challenge.
The multi-model approach tied closely to intensive measurements provides insights into the capability of
models to simulate background $O_3$ and harnesses the strengths of individual models to characterize the
sources of high-$O_3$ events. Stratospheric intrusions, Asian pollution, and wildfires are important sources
of the observed high-$O_3$ events above 65 ppbv in the SWUS, although uncertainties remain in the
quantitative attribution. These uncertainties may lie not only in $O_3$ sources but also in $O_3$ sinks, such as
removal by vegetation (e.g., Lin et al., 2019; 2020). Surface ozone in China continues to increase despite
regional $NO_x$ emission controls in recent years (Liu et al., 2016; Li et al., 2019; Sun et al., 2016).
Furthermore, the increasing frequency of wildfires under a warming climate (e.g., Westerling et al., 2006;
Dennison et al., 2014) and growing global methane levels (e.g., West et al., 2006; Morgenstern et al., 2013)
may foster higher background $O_3$ levels in the coming decades (Lin et al., 2017). These increasing
background $O_3$ sources, together with year-to-year variability in stratospheric influence (Lin et al., 2015a),
will leave little margin for $O_3$ produced from local and regional emissions, posing challenges to achieving
a potentially tightened $O_3$ NAAQS in the SWUS.

*Data availability.* Model simulations presented in this manuscript are available upon request to the
corresponding author (Meiyun.Lin@noaa.gov). Field measurements during *FAST*-LVOS are available at
https://www.esrl.noaa.gov/csd/projects/*FAST*lvos.
*Author contributions.* MYL conceived this study and designed the model experiments; LZ performed the
GFDL-AM4 simulations and all analysis under the supervision of MYL; EK and YXW conducted the
GEOS-Chem simulations; LWH and YXW assisted in the interpretation of model results; AOL, CJS, RJA,
IP, PC, JP, TBR, SSB, ZCJD, GK, and SC carried out field measurements. LZ and MYL wrote the article
with inputs from all coauthors.
*Competing interests.* The authors declare that they have no conflict of interest.
*Disclaimer.* The statements, findings, and conclusions are those of the author(s) and should not be
construed as the views of the agencies.
*Acknowledgements.* This work was funded by the Clark County Department of Air Quality (CCDAQ)
under contracts CBE 604279-16 (Princeton University), CBE 604318-16 (NOAA ESRL), and CBE
604380-17 (Scientific Aviation). MYL and LZ were also supported by Princeton University's Cooperative
Institute for Modeling the Earth Science (CIMES) under awards NA14OAR4320106 and

NA18OAR4320123 from the National Oceanic and Atmospheric Administration, U.S. Department of Commerce. We are grateful to Zheng Li (CCDAQ), Songmiao Fan (GFDL) and Yuanyu Xie (Princeton University) for helpful discussions and suggestions. We thank Qiang Zhang (Tsinghua University) for providing trends of anthropogenic $NO_X$ emissions in China and Christine Wiedinmyer (University of Colorado) for the 2017 FINN emission data.

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

**Table 1.** List of high-$O_3$ events above 65 ppbv in the greater Las Vegas region during April-June 2017 (unit: ppbv).

| Events | MDA8 $O_3$ (1-min max) at Angel Peak | Simulated MDA8 $O_3$ (USB) at Angel Peak: AM4 vs. GC | MDA8 $O_3$ at Clark County sites | Maximum MDA8 $O_3$ at rural sites in affected regions | Observed $\Delta O_3/\Delta CO$ | Observed $H_2O$ (g/kg) | Vertical profiles; synoptic maps | Surface impacts |
|---|---|---|---|---|---|---|---|---|
| **Stratospheric intrusions** | | | | | | | | |
| April 22-23 | - | 66 vs. 53 (60 vs. 47) | SM Youth Camp: 70; Green Valley: 67 | **Apr 22: WY:** Centennial (76); **CO:** Mesa Verde NP (72), Gothic (82) **Apr 23: WY:** Centennial (75); **CO:** Rocky Mt. NP (70); **CA:** Joshua Tree (76) | - | - | Fig. S6 | Figs. 4 and S6 |
| May 13-14 | - | 66 vs. 52 (62 vs. 48) | May 13: SM Youth Camp: 70; May 14: SM Youth Camp: 71 | **May 13: CA:** Joshua Tree (74); **UT:** AQS site: Zion NP (69) **May 14: NV:** Great Basin NP (65) | - | - | Fig. S6 | Figs. 4 and S6 |
| June 11-13 | June 11: 66 (84) | 65 vs. 47 (58 vs. 42) | Jun 11: SM Youth Camp: 64 | **Jun 12: WY:** Centennial (70); **CO:** Mesa Verde (69) **Jun 13: WY:** Centennial (65); **AZ:** Petrified Forest (65); AQS sites: Payson (76); **NM:** AQS sites: Cayote (71) | -3.79 | 0.6±0.2 | Figs. 7-8a | Fig. 9 |
| **Combined stratospheric and regional pollution influences** | | | | | | | | |
| June 14 | 73 (80) | 69 vs. 57 (53 vs. 50) | Joe Neal: 74 North LV Airport: 73, Walter Johnson: 71 | **CA:** Joshua Tree (95); **AZ:** Petrified Forest (71); **NM:** site: Bernalillo (71) | 0.75 | 2.5±0.3 | Fig. 8b | Fig. 9 |
| **Wildfires** | | | | | | | | |
| June 22 | 67 (83) | 58 vs. 76 (44 vs. 62) | Joe Neal: 78 North LV Airport: 82 | **CA:** Sequoia NP (86); Joshua Tree (74) | 0.015 | 3.5±0.2 | Fig. 10a and 11a | Fig. 12a |
| **Regional/local pollution events** | | | | | | | | |
| June 2 | 71 (78) | 61 vs. 64 (51 vs. 49) | Joe Neal: 66 Walter Johnson: 69 | **CA:** Joshua Tree (68, Jun 1: 79) | 1.09 | 2.8±0.3 | Fig. S10 | Fig. S11 |
| June 16 | 72 (82) | 65 vs. 63 (46 vs. 54) | Joe Neal: 75 Palo Verde: 75 | **CA:** Joshua Tree (98); **AZ:** Petrified Forest (65), AQS site: Payson (76) | 0.68-0.70 | 2.6±0.4 | Fig. 11b | Fig. 12b |
| June 29-30 | June 29: 71 (78) June 30: 75 (86) | 55 vs. 62 (41 vs. 54) | Jun 29: Joe Neal: 70; North LV Airport: 74 Jun 30: Joe Neal: 75; Walter Johnson: 75 | **Jun 29**: **CA:** Sequoia NP (74); Joshua Tree (75) **Jun 30**: **CA:** Sequoia NP (83); Joshua Tree (96); **AZ:** Grand Canyon (66) | 0.69-1.07 | 2.8±0.3 | Fig. S10 | Fig. S11 |
| **Long-range transport of Asian pollution; possibly mixed with local pollution** | | | | | | | | |
| May 24 | 65 (74) | 62 vs. 68 (48 vs. 54) | Arden Peak: 72, SM Youth Camp: 66, Jean: 66, Palo Verde: 65 | **CA:** Yosemite NP (70); **ID:** AQS site: Butte (69); **WY:** Yellowstone NP (64); **UT:** AQS site: Zion NP (65) | - | - | Figs. 13-14a | Fig. 15a |
| **Unattributed event** | | | | | | | | |
| June 28 | 68 (84) | 53 vs. 59 (43 vs. 54) | Joe Neal: 75; North LV Airport: 74 | **CA:** Sequoia NP (70); **AZ:** Grand Canyon (66) | 1.92 | 1.7±0.1 | Fig. 10b | Fig. 15b |

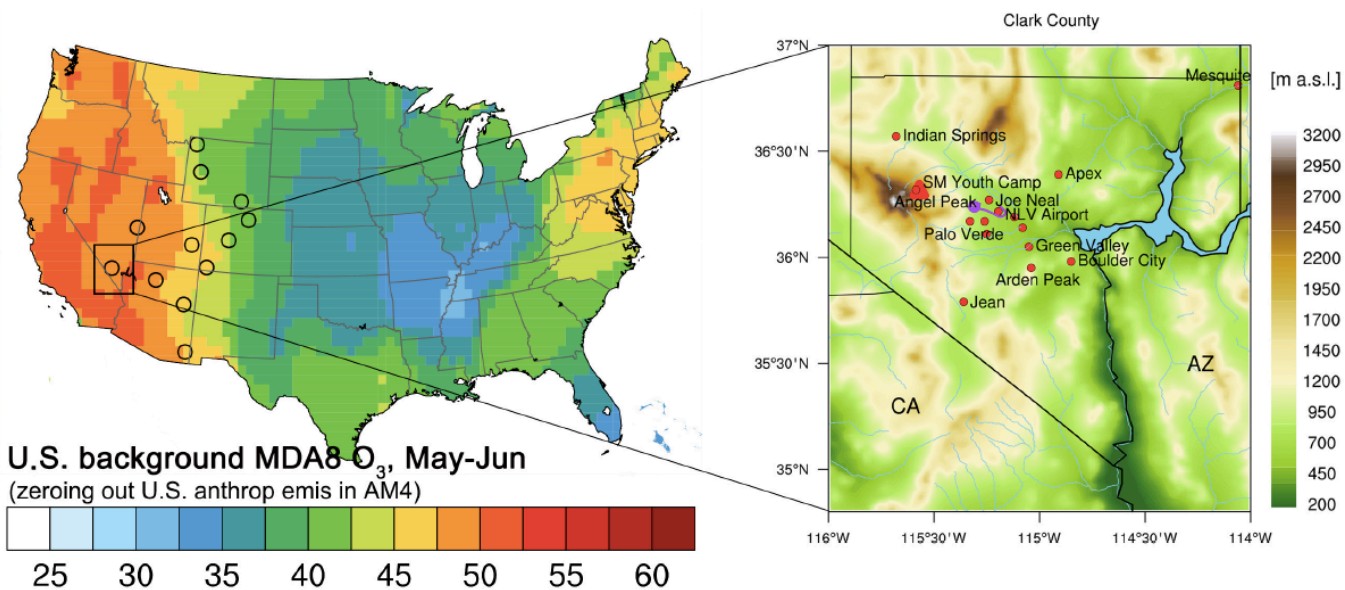

**Figure 1.** (Left) Mean U.S. background MDA8 $O_3$ (ppbv) during *FAST*-LVOS (May–June, 2017) estimated by zeroing out U.S. anthropogenic emissions in the global high-resolution (~50 km × 50 km) version of the GFDL-AM4 model (circles denote 12 selected high-elevation CASTNet sites); (Right) Topographic map of Clark County displaying the locations of Angel Peak (filled triangle) and regulatory $O_3$ monitoring sites (filled circles). The purple trace denotes the Scientific Aviation flight track during 19:15-19:35 UTC of June 28, 2017. The topographic data is from NOAA's National Centers for Environmental Information (http://www.ngdc.noaa.gov/mgg/global).

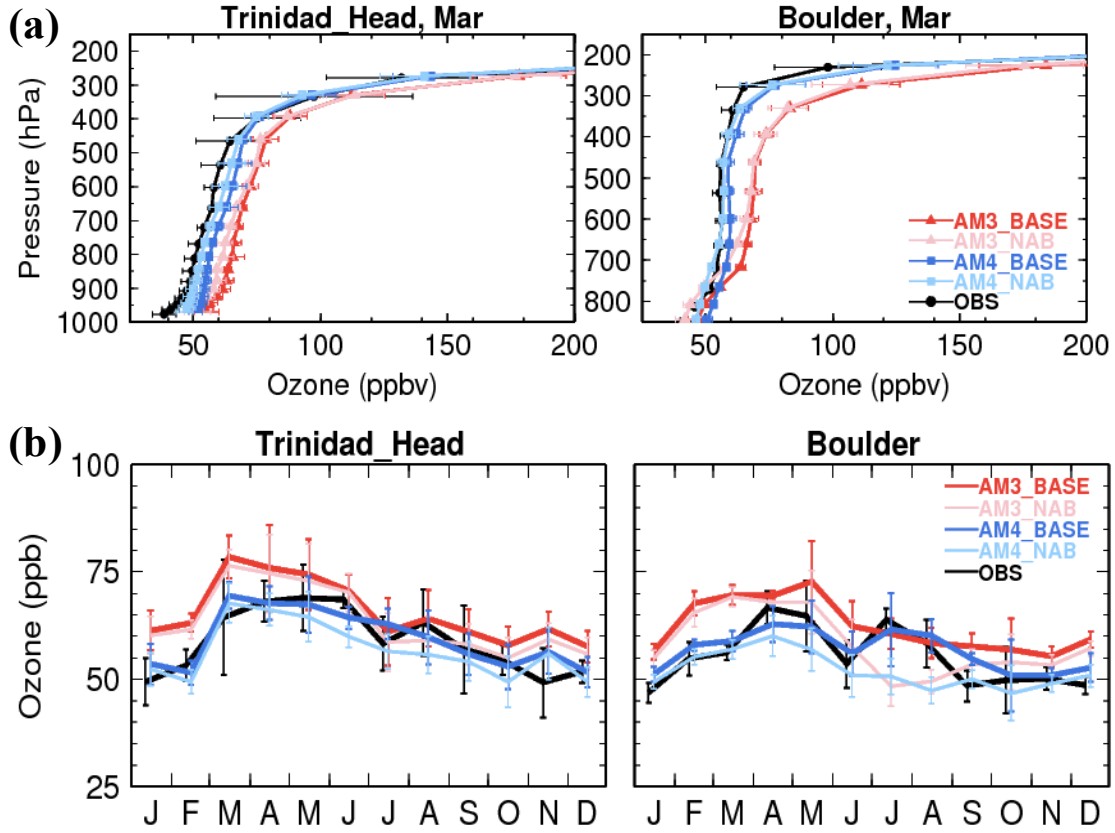

**Figure 2.** (a) Vertical profiles of O₃ in March and (b) monthly mean O₃ in the middle troposphere (500–430 hPa) at Trinidad Head, California (41.1°N, 124.2°W, 107 m a.s.l.) and Boulder, Colorado (40.0°N, 105.0°W, 1584 m a.s.l.) during 2010–2014 as observed (black) and simulated by GFDL-AM3 (red; AM3_BASE; Lin et al., 2017) and GFDL-AM4 (blue; AM4_BASE), together with simulated North American Background O₃ (NAB; estimated with North American anthropogenic emissions zeroed out). The bars represent the standard deviations of monthly values during 2010–2014.

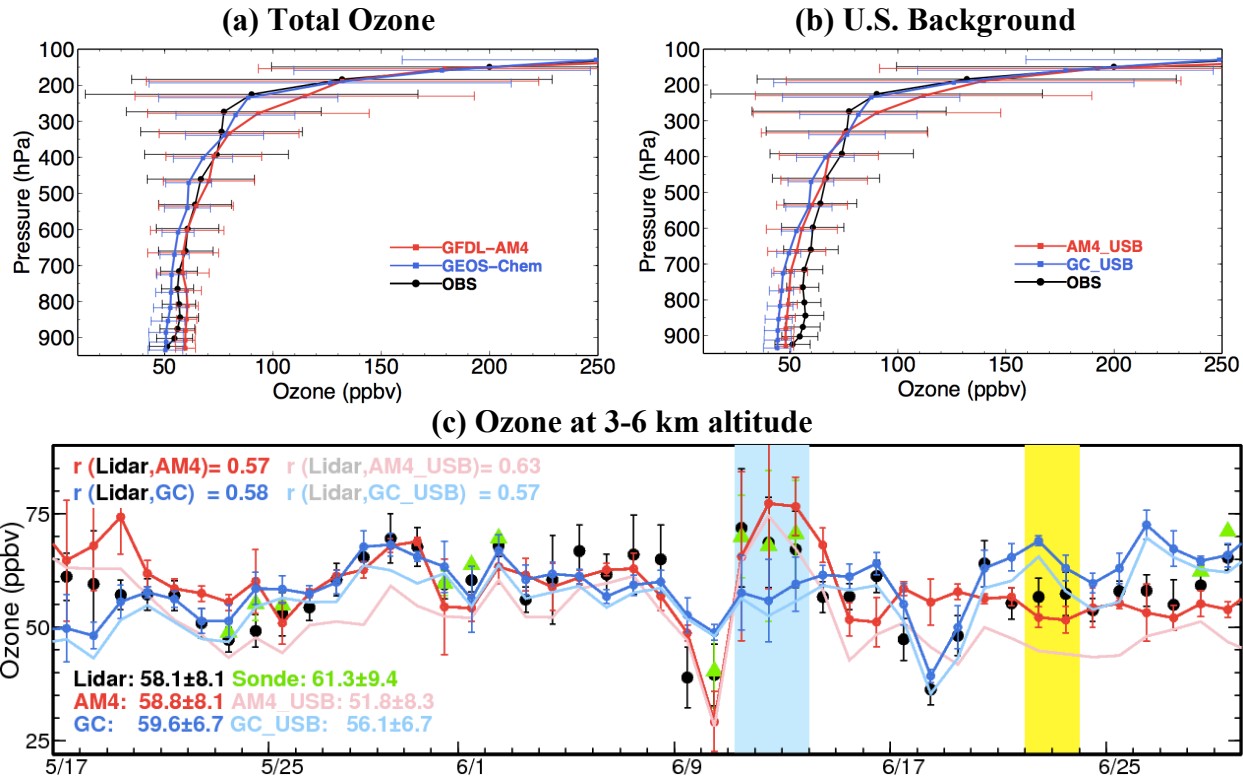

**Figure 3.** (a) Mean vertical O₃ profiles at Joe Neal as observed with ozonesondes (black; 30 launches) and simulated with GFDL-AM4 (red) and GEOS-Chem (blue) during *FAST*-LVOS (May–June 2017). Horizontal bars represent the standard deviations across daily profiles; (b) Same as (a), but showing U.S. background (USB) O₃ estimated by the two models. (c) Time series of O₃ averaged over 3-6 km altitude above NLVA during FAST-LVOS as observed (black: lidar; green: ozonesonde) and simulated with GFDL-AM4 (thick red line) and GEOS-Chem (thick dark blue line), together with simulated USB O₃ (light lines). Here and in other figures, AM4_USB represents USB estimated by GFDL-AM4 and GC_USB represents USB estimated by GEOS-Chem. The blue shading highlights the period with stratospheric intrusions and the yellow shading, the wildfire event. Vertical bars represent the standard deviations across hourly averages.

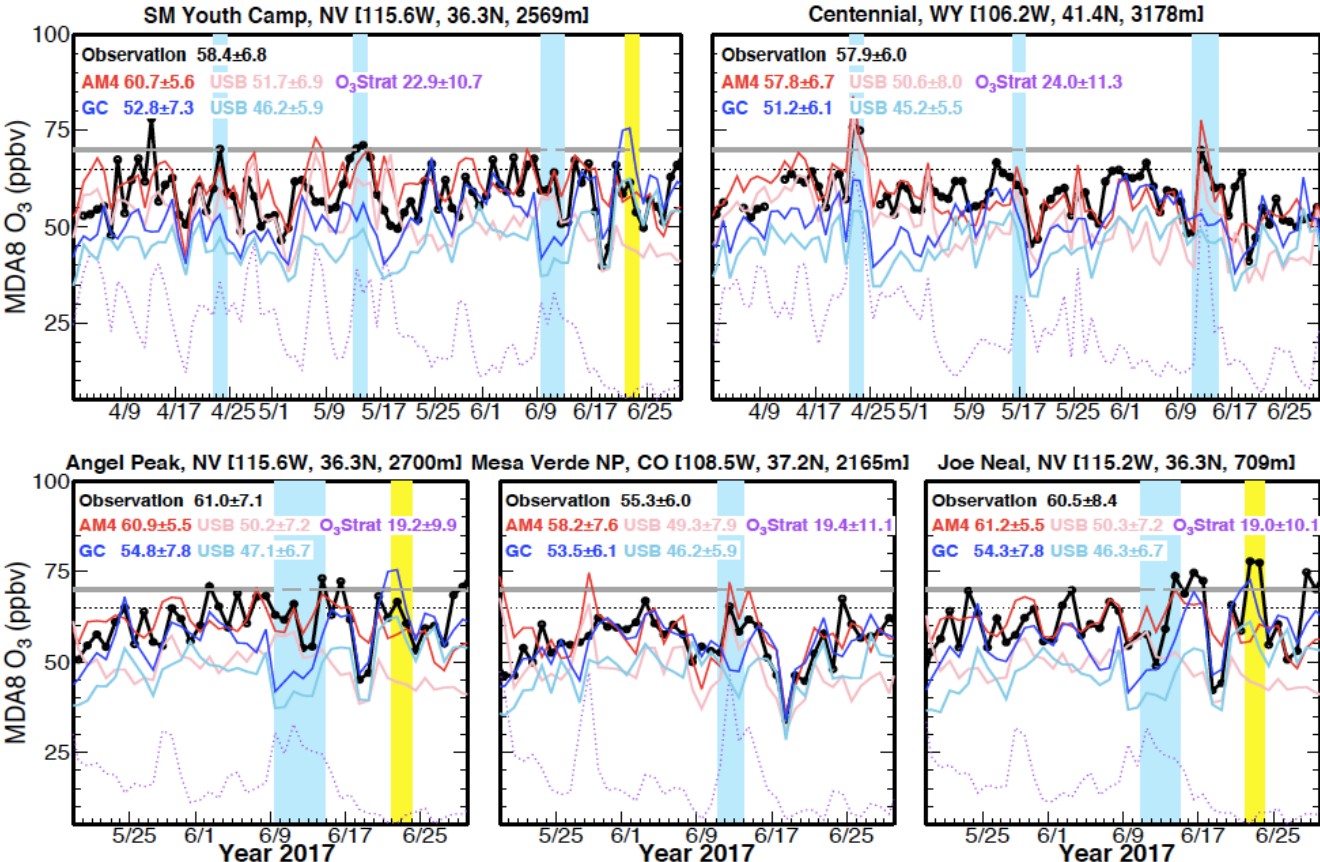

**Figure 4**. Time series of daily MDA8 O$_3$ at Spring Mountain Youth Camp (SMYC) in Nevada and Centennial in Wyoming from April to June, and at Angel Peak, Mesa Verde, and Joe Neal during the *FAST*-LVOS study period, highlighting stratospheric intrusion events (blue shading) and wildfire events (yellow shading). The SMYC O$_3$ monitor is located only about 125 m below, and 800 m west of the Angel Peak summit where the mobile lab was parked. Shown are total MDA8 O$_3$ from observations (black) and simulations by GFDL-AM4 (red) and GEOS-Chem (blue), together with USB O$_3$ from GFDL-AM4 (pink) and GEOS-Chem (light blue). The dashed purple line shows AM4 stratospheric O$_3$ tracers The horizontal lines denote the current NAAQS level of 70 ppbv and a possible future standard of 65 ppbv.

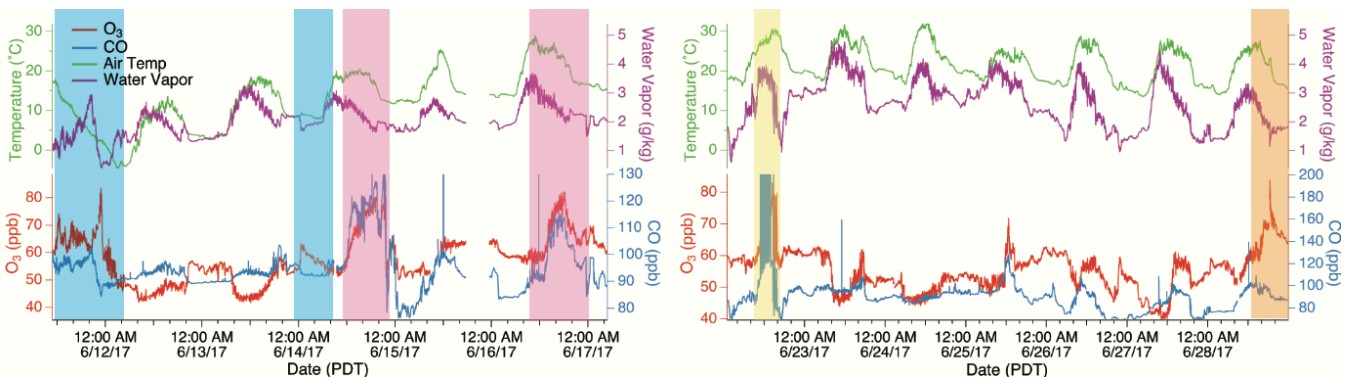

**Figure 5.** Time series of 1-minute averaged air temperature, water vapor, O₃, and CO mixing ratios measured by the NOAA mobile lab deployed at Angel Peak during June 11−16 and June 22−28, 2017, highlighting the periods with stratospheric influence (blue), regional anthropogenic pollution plumes (pink), wildfire plumes (yellow), and the unattributed pollution plume (orange). Data are shown in Pacific Daylight Time (PDT). Note that peak CO mixing ratios on June 22 were 440 ppbv (not shown on the plot).

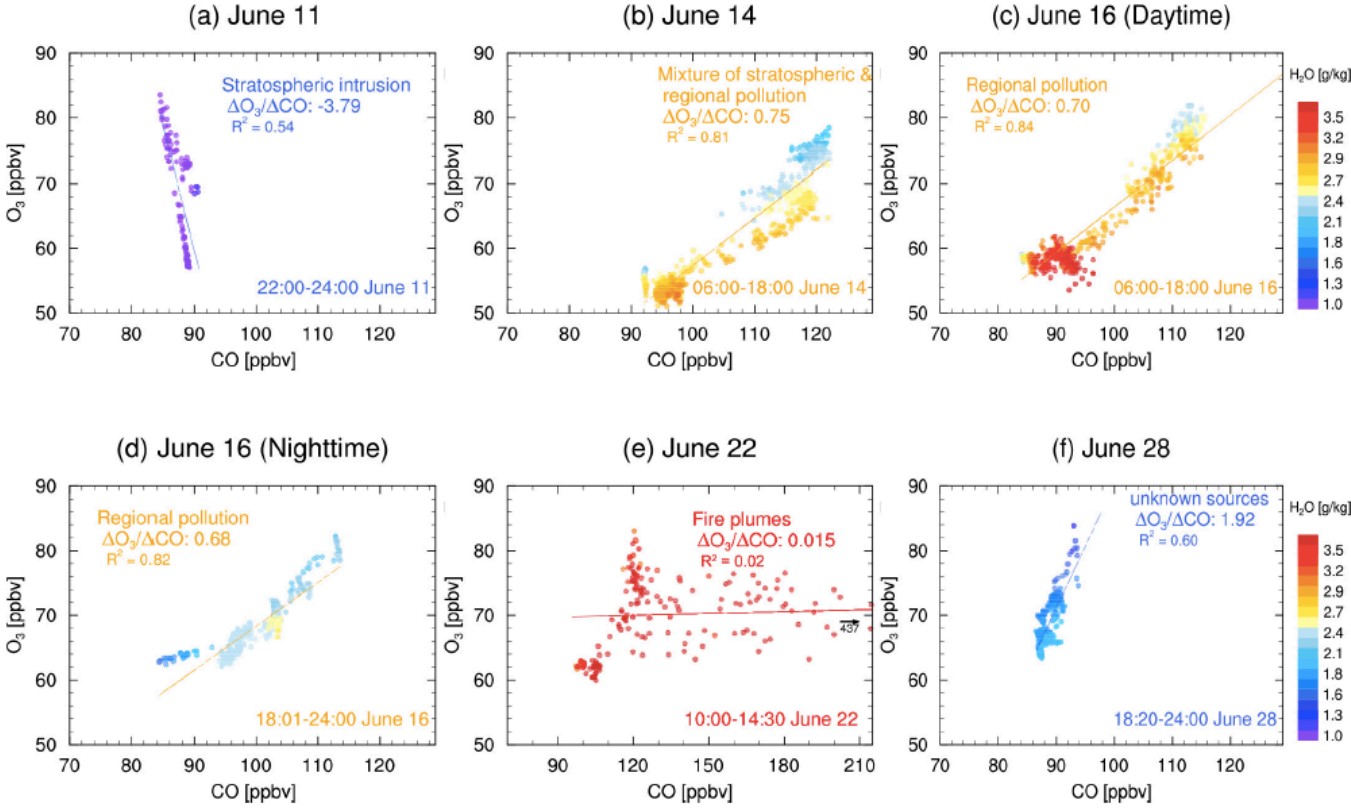

**Figure 6.** Scatter plots of 1-min average $O_3$ against CO measured at Angel Peak, color-coded by specific humidity, for air masses influenced by (a) STT on June 11; (b) regional pollution on June 14; (c–d) regional pollution plume during daytime (06:00–18:00) and nighttime (18:01–24:00) on June 16; (e) wildfires on June 22; and (f) unattributed pollution on June 28. Note that peak CO mixing ratios on June 22 were 440 ppbv (not shown on the plot).

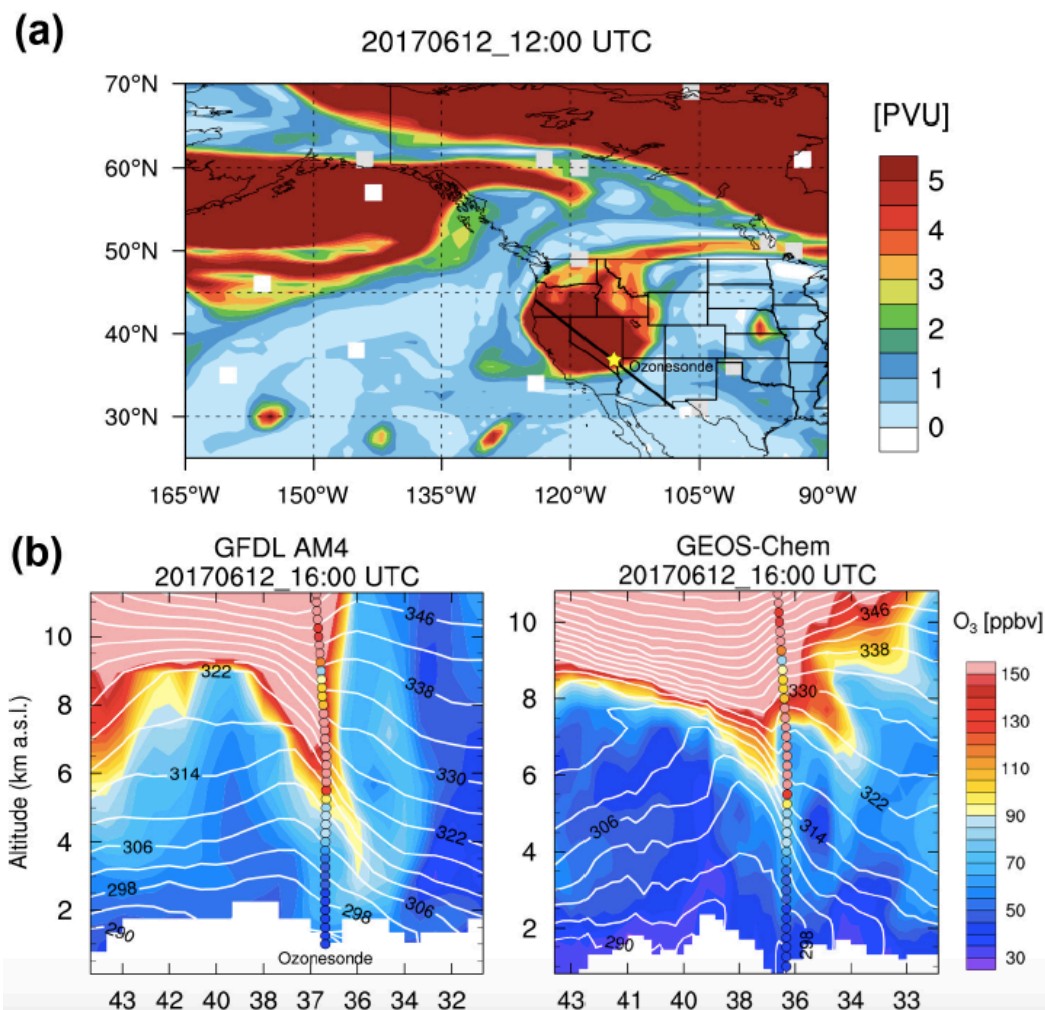

**Figure 7.** (a) Potential vorticity at 250 hPa on June 12 calculated from the NCEP-FNL reanalysis (PVU: $10^{-6}$ m$^2$ s$^{-1}$ K kg$^{-1}$); (b) vertical distributions of O$_3$ (color shading) and isentropic surfaces (white lines) along a transect crossing Nevada (black line on PV map) simulated with GFDL-AM4 (left) and GEOS-Chem (right) on June 12. The color-coded circles denote ozonesonde observations at Joe Neal (star on the PV map).

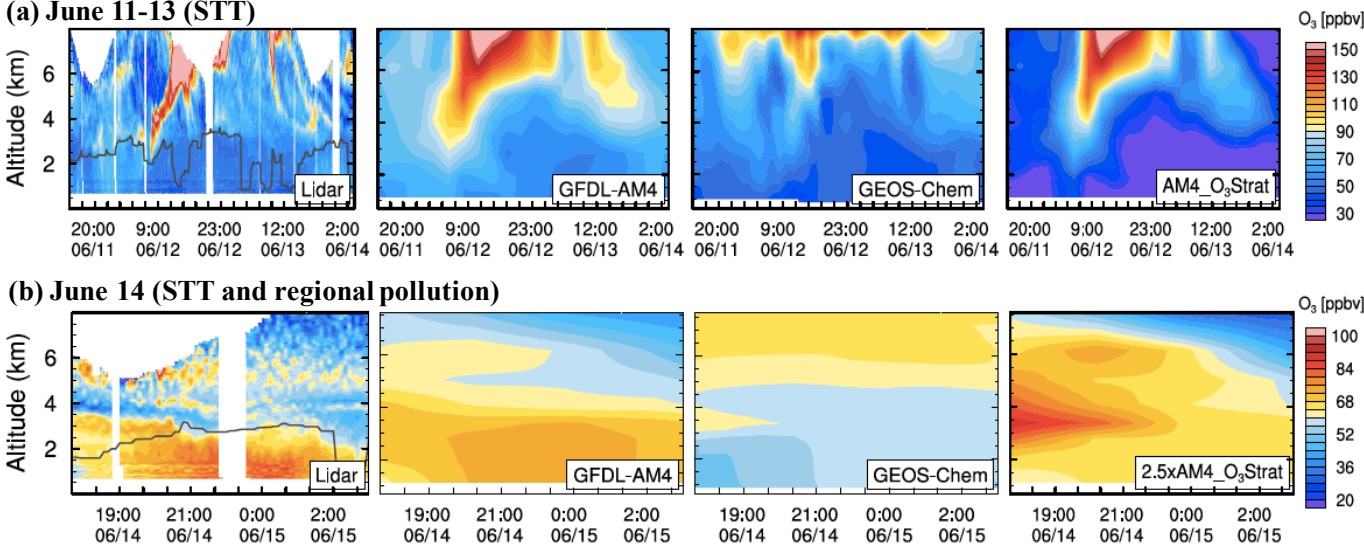

**Figure 8.** Time-height curtain plots of $O_3$ above NLVA as observed with TOPAZ lidar and simulated with GFDL-AM4 (~50 km × 50 km; interpolated from 3-hourly data) and GEOS-Chem (0.25° × 0.3125°; interpolated from hourly data) during the STT event on (a) June 11−13 and (b) June 14, 2017 (UTC). The rightmost panel shows AM4 stratospheric $O_3$ tracer (AM4_$O_3$Strat). Note that AM4 $O_3$Strat for June 14 is scaled by a factor of 2.5 for clarity. Here and in other figures, the solid black lines in the $O_3$ lidar plots represent boundary layer height inferred from the micro-Doppler lidar measurements.

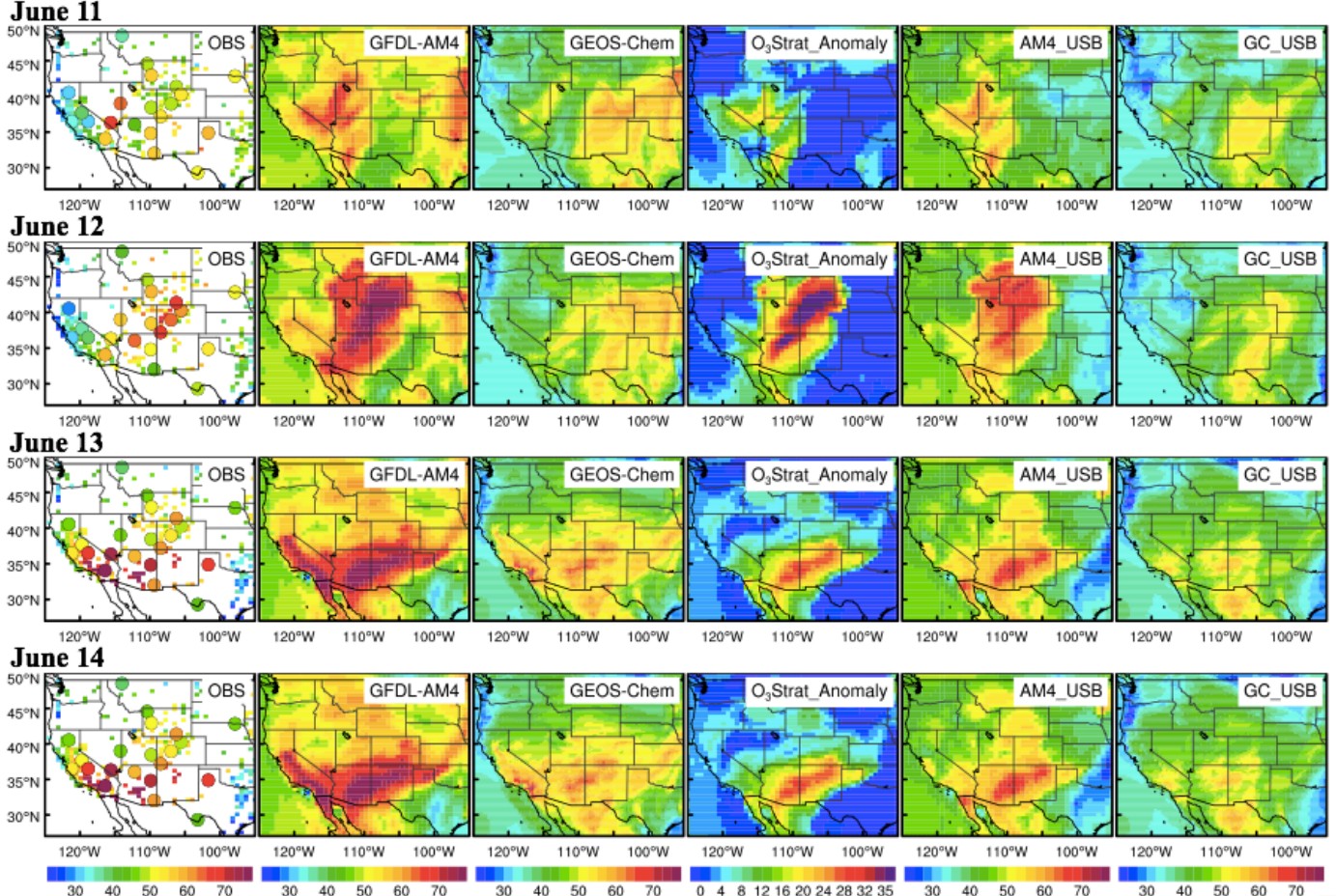

**Figure 9.** Maps of total MDA8 O₃ (ppbv) in surface air as observed (small squares for AQS data and large circles for CASTNet data) and simulated with GFDL-AM4 and GEOS-Chem, along with anomalies in AM4 O₃Strat (relative to June mean) and model-estimated USB levels, during the STT event on June 11–14, 2017. Note that O₃Strat in this figure and Fig.S6 is shown as anomalies relative to the monthly mean, while the absolute values are shown in Figs.4 and 8.

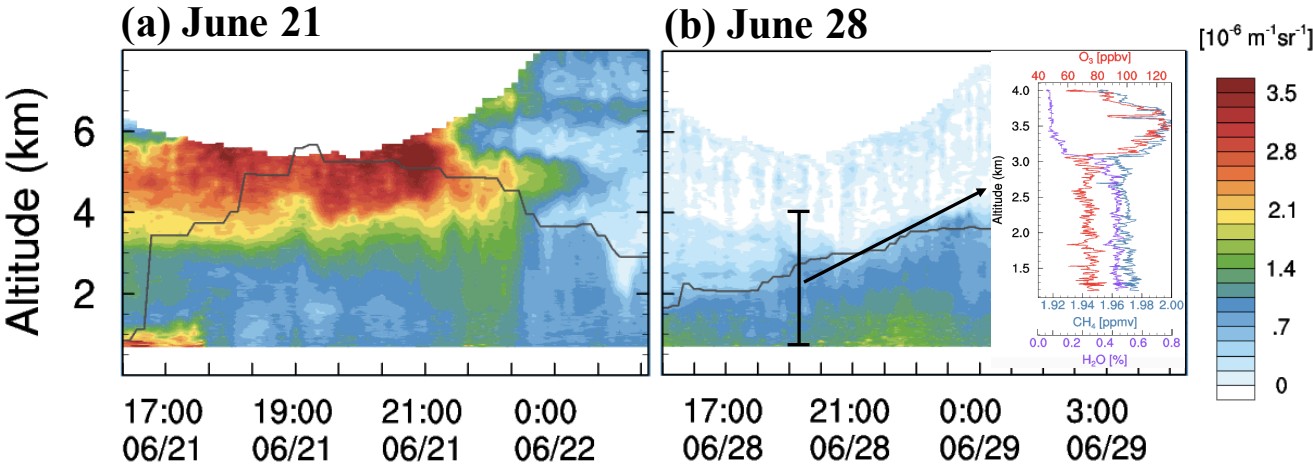

**Figure 10.** Time-height curtain plots of the TOPAZ aerosol backscatter above the North Las Vegas Airport during June 21–22 (a) and June 28, 2017 (b). Data are shown at UTC time. The inset graph in (b) shows vertical profiles of water vapor (purple), $CH_4$ (blue), and $O_3$ (red) measured by the Scientific Aviation flight above the Las Vegas Valley during 19:15–19:35 June 28 (UTC) (flight track in Figure 1).

**(a) June 22 (wildfires)**

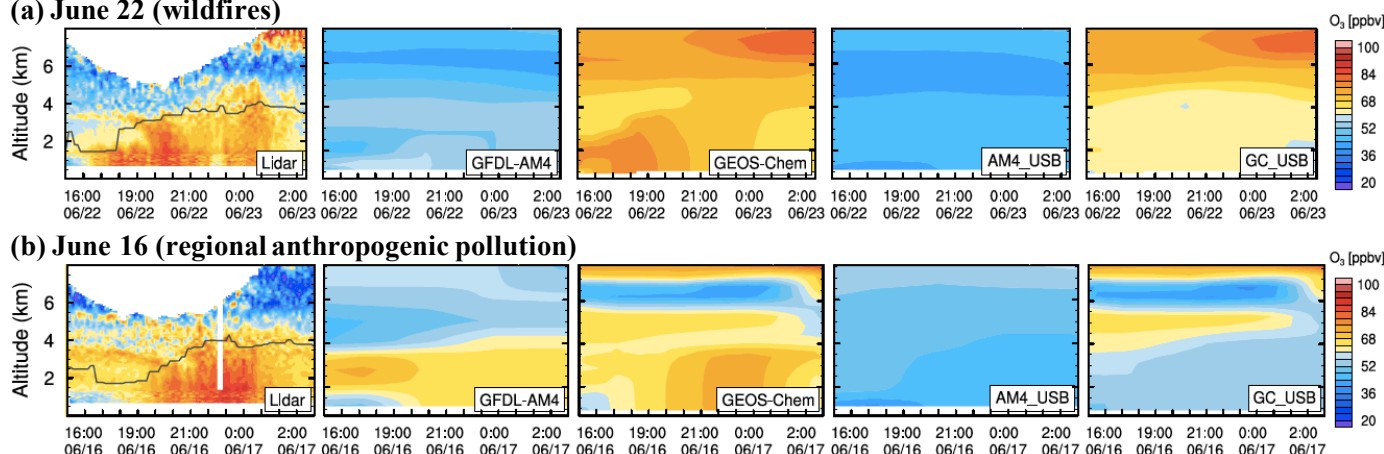

**(b) June 16 (regional anthropogenic pollution)**

**Figure 11.** Same as Figure 8, but for (a) the wildfire event on June 22 and (b) the regional anthropogenic pollution event on June 16, 2017 (UTC). The right panels compare USB O₃ from the two models.

**(a) June 22 (wildfires)**

**(b) June 16 (regional anthropogenic pollution)**

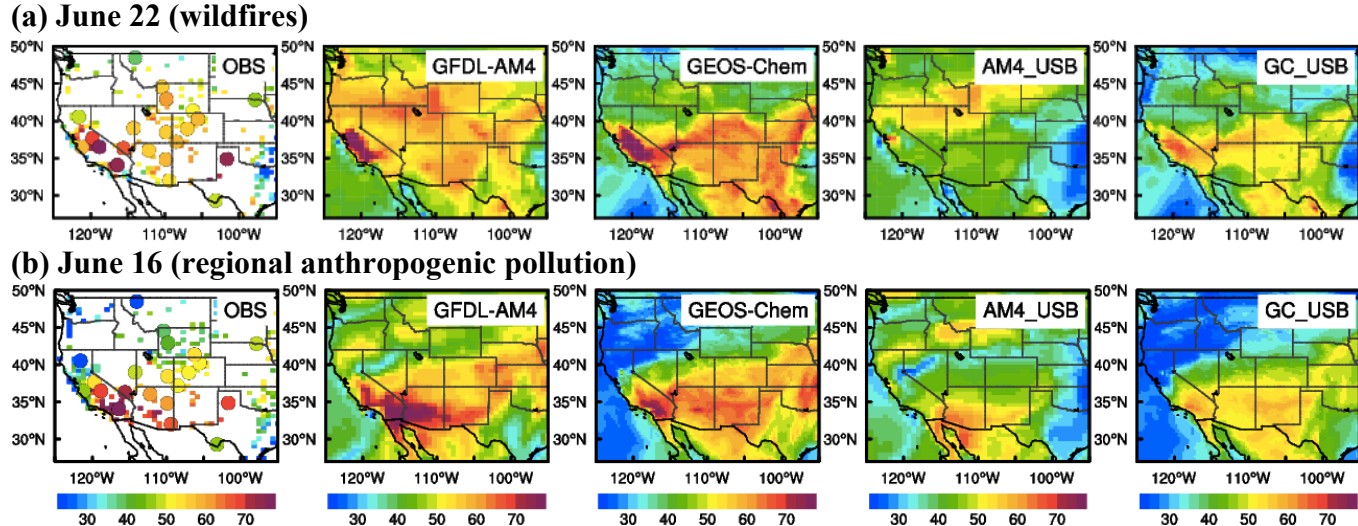

**Figure 12.** Same as Figure 9, but for (a) the wildfire event on June 22 and (b) the regional anthropogenic pollution event on June 16, 2017.

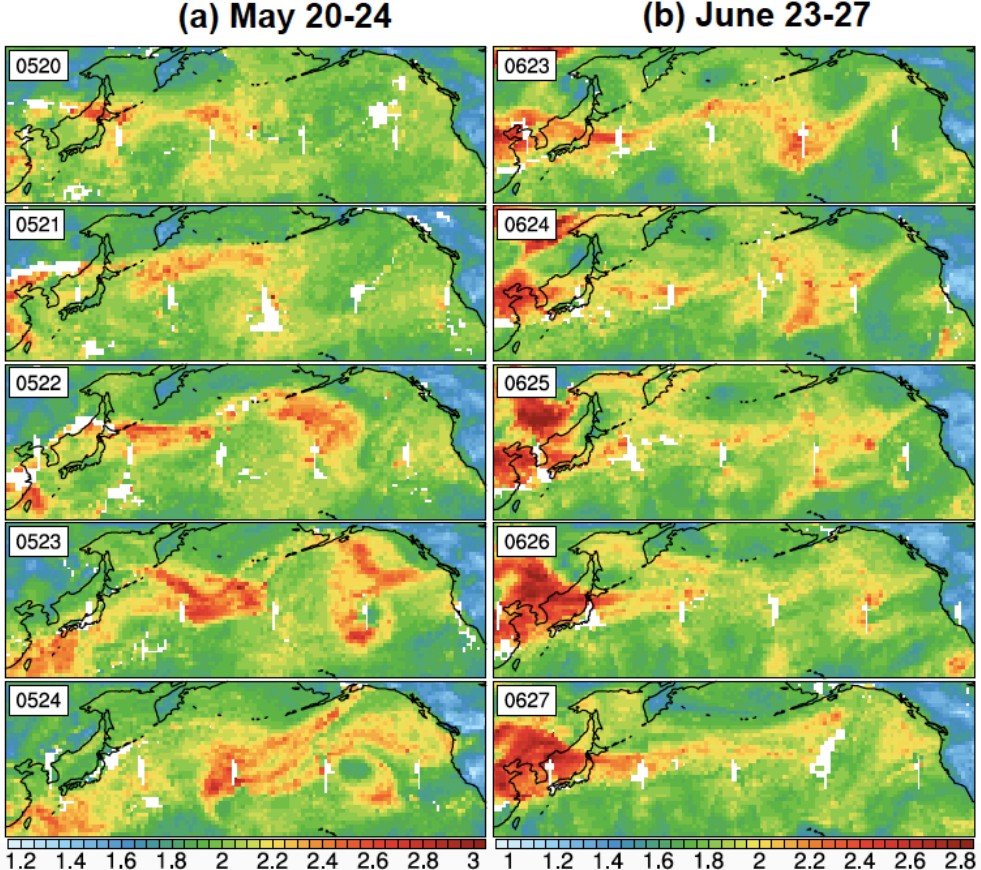

**Figure 13.** Trans-Pacific transport of Asian pollution plumes during (a) May 20–24 and (b) June 23–27, 2017, as seen in the NASA AIRS retrievals of CO total column ($10^{18}$ molecules/cm$^2$; level 3 daily 1°×1° gridded products).

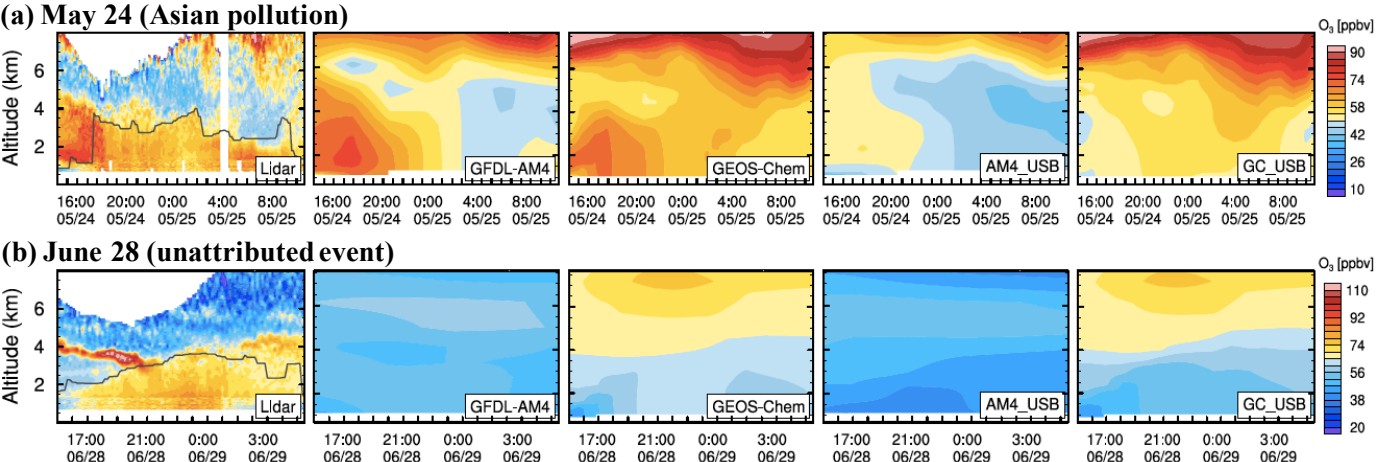

**Figure 14.** Same as Figure 8, but for (a) the Asian pollution event on May 24 and (b) the unattributed pollution event on June 28, 2017 (UTC).

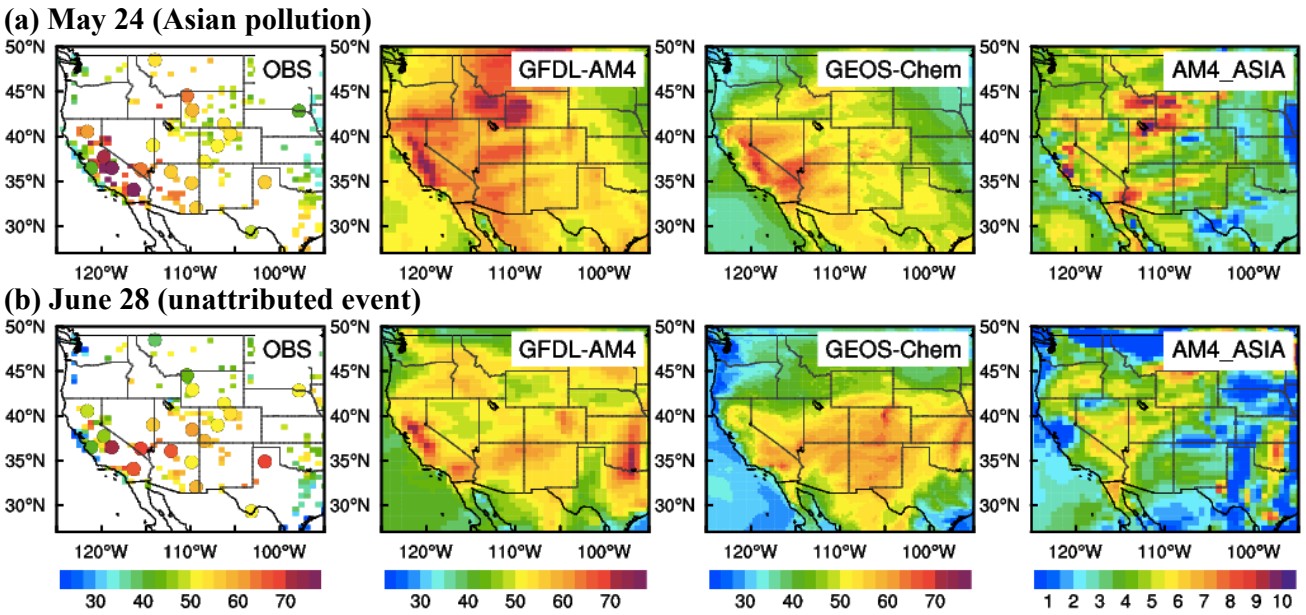

**Figure 15.** Same as Figure 9, but for (a) the Asian pollution event on May 24 and (b) the unattributed pollution event on June 28, 2017. The right panels show $O_3$ enhancements from Asian pollution estimated by GFDL-AM4.

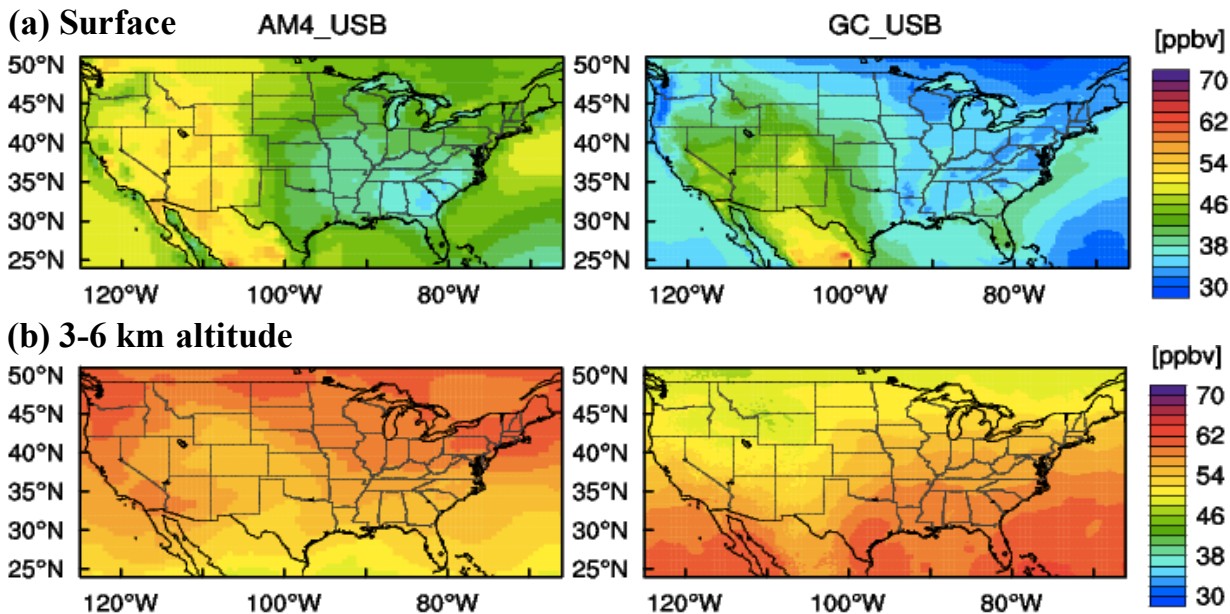

**Figure 16.** Spatial distributions of USB O₃ simulated with GFDL-AM4 and GEOS-Chem (a) at the surface (MDA8) and (b) at 3−6 km altitude (24-hour mean) during April−June, 2017.

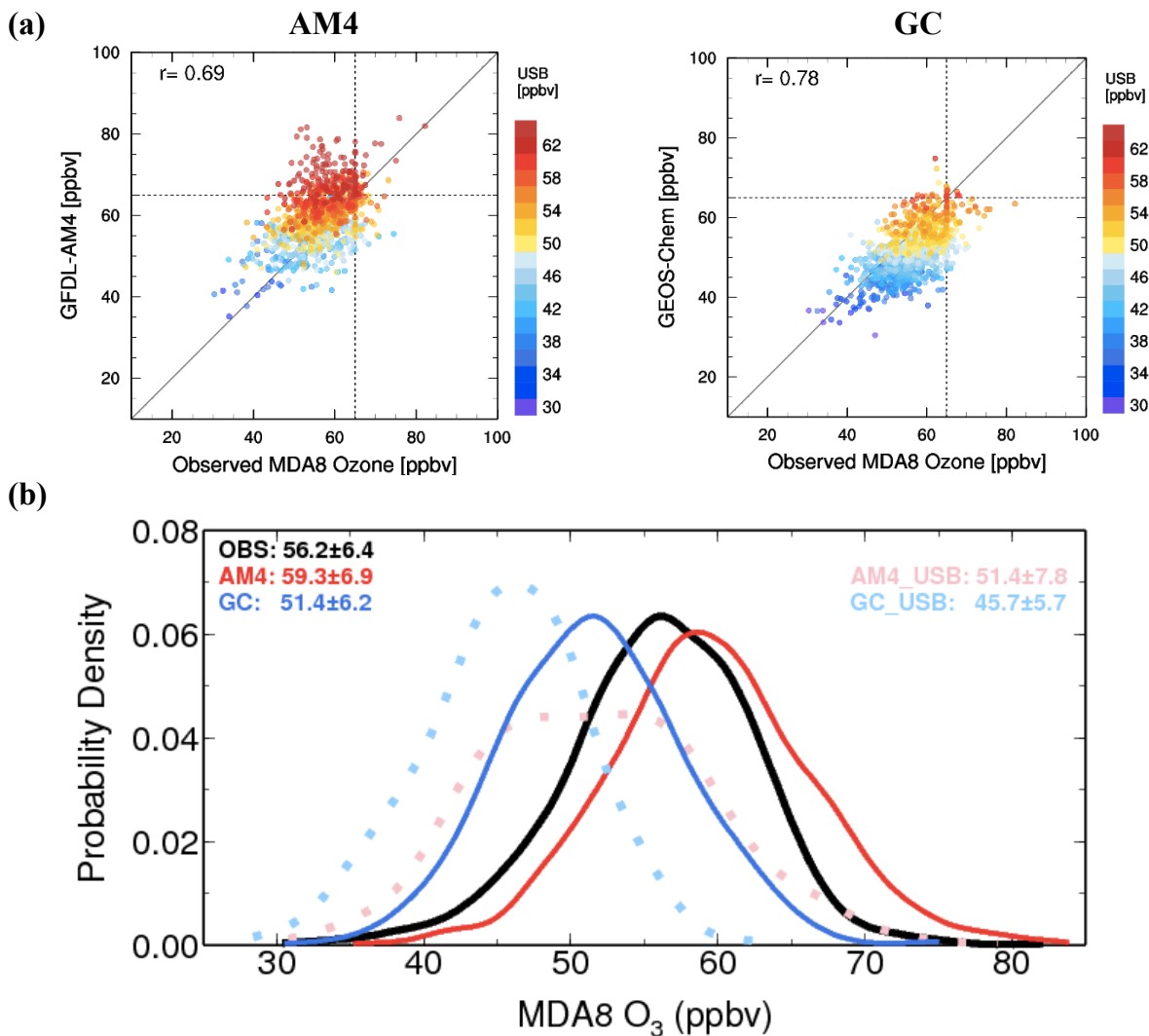

**Figure 17.** (a) Scatter plots of observed versus simulated daily MDA8 O₃, color-coded by USB O₃, at 12 WUS high-elevation sites (circles in Figure 1a) during April–June, 2017. The dashed lines mark the 65-ppbv threshold; (b) Probability density of daily MDA8 O₃ as observed (solid black) and simulated with GFDL-AM4 (solid red) and GEOS-Chem (solid blue), along with the distribution of USB O₃ estimated from each model (dotted lines).