# Peer review of "Characterizing sources of high surface ozone events in the southwestern U.S. with intensive field measurements and two global models"

_Atmospheric Chemistry and Physics, 2019_

## Referee Comment (RC1) · Anonymous Referee #1 · 6 Jan 2020

In this article, Zhang et al evaluate ambient and modeled data in the Las Vegas area during the 2017 FAST-LVOS study. They aim to determine dominant source categories for high observed ozone events. They leverage both enhanced monitoring data and sensitivity simulations from multiple global models to understand O3 events at this time and location. The evaluations shown here highlight the challenges with determining O3 sources. Even with these detailed datasets the evidence for categorizing O3 events on many of the event days is not definitive. I think this analysis is a valuable addition to the literature, but I think the uncertainties in the analysis need to be more clearly communicated. I believe that the authors have overstated the confidence in their ability to categorize ozone events from the data provided. In addition, there are certain areas where the article needs additional details and background, specifically: 1) include references to additional relevant articles on background ozone, 2) include more comprehensive model evaluation information especially as it relates to the specific times and locations of the ozone events of interest, 3) provide more systematic information with which to compare indicator values in each these events through Table 1 or additional tables/figures.

Major comments:

1. Introduction: Please reference the recent comprehensive review of background ozone by Jaffe et al (2018). For the paragraph summarizing past modeling to predict USB, please also include references to Dolwick et al (2015), Emery et al (2012) and additional references cited in Jaffe et al (2018). In lines 80-85, please note that the Dolwick et al (2015) modeling included analysis of daily O3 and plotted USB against total O3 showing the range of daily values. Jaffe et al (2018) also included daily quantification of background ozone. Similarly, when discussing past literature on USB estimates in section 5, please also cite and compare to Dolwick et al (2015) and Emery et al (2012).

2. Modeling description and evaluation:

- Given the heavy reliance on AM4 results to categorize events in this paper, the authors should provide more detailed information on model performance. Model performance is only shown for aloft measurements. I suggest adding model performance of ground-level O3 based on measurements at CASTNET sites. Also, it would be useful to report mean bias of ground-level O3 (based on nearby CASTNET and FAST-LVOS measurement) for each specific episode day (i.e. what is the model performance at the times and locations of interest and how does it change on different days examined). This will provide important context for interpreting modeling results used to classify the different O3 episodes. This could be done as part of Table 1 or as a separate table in the paper. In addition, performance information that is already available in Figures 15

and 17 could be brought forward and expanded upon in section 3.

-Given 50 km resolution of AM4, the model may be better suited to quantify O3 from some sources than from others. The coarse resolution may not matter as much for stratospheric intrusions and for transport from Asia but may be insufficient for capturing photochemical production from sources that may have gradients in precursor emissions (fires and local/regional US anthropogenic sources).

-The AM4 model simulations were conducted for January-June 2017 and the first episode evaluated was in late April. That corresponds to ∼4-month spin-up period which seems short for global simulations that are tracking impact from long-range sources. Please address this short spin-up period? Why is this length of spin-up appropriate for the simulations conducted here?

-A stratospheric tracer is implemented in AM4. Please provide details on whether this tracer is inert or reactive (i.e. can be degraded by chemistry and deposition). If the tracer is inert, then it should not be used to quantify stratospheric impacts (e.g. line 313) because the lack of degredation processes will lead to an overestimate of stratospheric O3 influence. In this case it could still be used qualitatively to identify times and locations of stratospheric influence.

-GEOS-Chem description did not specify the simulation period. Was this also Jan-June 2017? If so, please also address the relatively short spin-up on the GEOS-Chem simulations.

-I have several questions/comments on the US emissions adjustment. You state that NOx emissions were cut by 50% in the Eastern US, does this mean the NOx emissions were left unchanged in the Wester US? The Travis et al paper was based on an analysis for 2013 and NOx emissions and vehicle fleet characteristics (age, vehicle emissions control systems etc) have been continually changing in the US. Is it appropriate to apply scaling factors based on a 2013 analysis to this 2017 time period? In addition, GEOS-Chem emissions already account for decreasing NOx emissions in more recent

years based on EPA trends information. Since the EPA trends include improvements to inputs for mobile source emissions calculations in more recent years, this 50% adjustment may be double-correcting for adjustments that are already included in the more recent EPA data. In addition, recent papers coming out of the 2017 WINTER campaign (Salmon et al., 2018; Jaegle et al. 2018) suggested that EPA NOx emissions were unbiased in winter, so should same 50% NOx cut be applied in winter months as in summer? If this adjustment was only made in the Eastern US, perhaps the impact of these adjustment are limited for this analysis which focuses on the Las Vegas area. Also, why do you use monthly climatology for lightning NO rather than a method based on NLDN? Perhaps the impact of lightning NO representation on model performance noted on Line 387 would be less if actual emissions rather than monthly climatology were used. It would be useful to add a table to the supplemental information that included emissions levels (tons of NOx, CO, VOC) by region (Eastern US, Western US, China, EU, Fires etc) used for the 2 models.

3. Source characterization for specific O3 events:

For many of the episodes listed in table 1, the evidence is suggestive but not compelling for the classifications given.

I suggest adding to Table 1 a characterization of how confident you are in the classification. Based on evidence presented in the paper, the only episode that I would rate a "high confidence" is the June 11th stratospheric intrusion.

I suggest adding fields to Table 1 so that episodes can be more easily compared: H2O mixing ratio (Avg +/- SD), O3:CO slope, O3/NOz. Several of these indicators are mentioned for one of the episode days in the text but not provided for all days so that the reader can compare what they look like during different types of events.

It would be useful if Table 1 provided a more detailed accounting of which lines of evidence were used to classify each of the episodes examined. It seems that in most cases, the ambient data can be indicative of influence from different types of sources

but can't provide quantitative estimates of how much O3 is from US sources versus fires/stratosphere/Asia etc. The model zero out simulations are often the basis for determining that O3 during an episode is primarily from one source. Given this dependence on the model, more detail on model methods and model performance compared to ground-based measurement on each of the episode days is warranted.

3.1. Even characterization for stratospheric intrusions:

-Apr 22-23 and May 13-14 - no details are provided to evaluate these events

-June 11-13 - anti-correlated O3:CO and very dry conditions are compelling for predominant stratospheric impact. I suggest adding this episode to figures S5 and S6 to show what the stratospheric tracer looks like for these days.

-June 14 - Evidence not compelling. O3:CO is positively correlated and slope is similar to slope from "US anthropogenic events". O3/NOz evidence not provided for any other events for comparison and reference from 1998 analysis may not be relevant to current conditions. Note that a recent paper by Henneman et al (2017) found that O3/NOz values have increased over time as US NOx levels have decrease leading to more efficient O3 production. They reported that for Atlanta, the O3/NOz increased from 5.4 in 2001 to 9.3 in 2011. Similarly, in urban Gulfport, FL the ratio increased from 11.1 in 2001 to 20.5 in 2011. So, the range of 1-7 based on Kleinman et al (2002) may not be relevant for current urban conditions in the US. H2O mixing ratio and O3:CO slope appear to be in same range as other events classified as from US anthropogenic influence. Vertical profiles and modeling impacts are not compelling; vertical profiles look similar to what I would expect for local formation with low overnight PBL concentrations due to titration, some mixing down of residual layer in the morning accompanied by increasing boundary layer O3 during the day. What does the stratospheric tracer look like on this day?

3.2. Event characterization for wildfires (June 22): The increased aerosol backscatter aloft on the previous day (Figure 14) is suggestive that wildfires may be advected to

this general location. The O3:CO and H2O mixing ratio is not convincing. Why are O3 and CO not correlated if they are both originating from the fire event? Also, the H2O mixing ratio looks like it is in the range of what was observed on June 16th, a "US anthropogenic emissions" event. What are typical H2O mixing ratios (average +/- SD) for Las Vegas during June? Is 3.5 g/kg really outside what you might get from normal meteorological variation? In fig 9 it is clear that the models cannot accurately capture O3 on this day, so it does not appear that modeling evidence should be used here leaving us with no way to quantify the impact of wildfires on the ground-level ozone. Do the fire sensitivities described on lines 394-406 impact the vertical profiles of O3? If you were to recreate Figure 9 using those sensitivities would the models be more able to capture O3 vertical profile?

3.3. Event characterization for regional/local pollution events:

-June 16th: O3 and CO are positively correlated suggesting that O3 formed from an emissions source that also emitted CO. This appears to be based primarily on modeled predictions of total O3 and USB. How does this compare to other events? What is the mean bias for ground-level O3 in Las Vegas on this day? Please state the level of elevated ozone aloft measured by TOPAZ (55-70 ppb?)

-June 2 and June 29-30: Not much information provided. Event characterization for Asian transport event (May 24): AIRS CO images from the days leading up to May 24 are suggestive of transport from Asia. No information is provided on O3:CO or H2O mixing ratio. Modeling from AM4 predicts 5-10 ppb influence from Asia. Is 5-10 ppb of O3 from Asia enough to classify this as "Asian transport" alone since O3 as > 70 ppb? Doesn't this suggest that there was at least one other major source? What is the MB for O3 at ground-level in Las Vegas on this day? Also note that the elevated O3 at 1-4 km altitude could also be local/regional O3 in the residual layer from previous days. The elevated O3 at 6-8 km is more definitively long-range transport. For this episode, I would say it is inconclusive. The models predict an Asian contribution, but the observations are not conclusive as they could also be showing local/regional

photochemical production over several days. The evidence from AIRS is suggestive but cannot provide any quantification of how much of the O3 comes from Asia.

3.4. Event characterization for unattributed event (June 28): Much of the evidence here seems to rule out rather than support any particular source. The elevated filament seems uncharacteristic of longer range transport over which time the filament is likely to disperse. Yet the satellite CO suggests possible transport form Asia. The back trajectories suggest possible influence from a California wildfire but the relatively dry air and lack of elevated aerosol backscatter makes it seem unlikely that a narrow filament of O3 could be from a wildfire without substantially elevated PM. The narrow filament of O3 could be from the stratosphere but the elevated CO and CH4 suggest photochemical formation. It should also be noted that Sequoia NP often receives pollution from California's San Joaquin Valley and Los Angeles, so the back trajectories make it seem equally likely that the source is US emissions vs a wildfire. How do the deltaO3/deltaCO and H2O mixing ratio compare to other events. Is the O3:CO slope statistically different from slopes on other days (using 95 % CO from the regression)? Could you use a T-test on the mixing ratio to determine whether it is statistically different from H2O levels on other episode days? One line 307, you indicate that the dryer air during this episode indicates the plume was from Asia or Los Angeles, which is a different conclusion than you present on lines 470-472 that states it is likely from a wildfire plume. What is the model mean bias (in ppb) for ground-level O3 on this day?

4. Conclusions: I think the language in the conclusion is too strong since there appears to be a lot of uncertainty in the characterization of O3 sources for most of the events evaluated in this paper. For instance, I do not believe that this analysis has been successful in "pinpointing sources of observed MDA8 O3" (line 536) and the language in general should reflect the uncertain nature of the conclusions that can be drawn for the analysis so far.

Other specific comments:

-Line 300: did you mean to indicate specific panels of Figure 5?

-Figure 6: suggest marking location of O3 sonde in panel a)

-Figure 7: Suggest using a separate color palette for the rightmost panel of AM4 rather than scaling by 2.5 to avoid confusion by the reader.

-Figure 15: Some but not all of these monitors are shown in the map in Figure 1. I suggest you provide a map that has locations of all monitors used in Figure 15.

-Figure 16: Why do you use 2 different scales in panels a) and b)? It looks like the O3 covers the same range of values in both. It is confusing to the reader when trying to compare the results from the various panels. I suggest just using the color range from panel a) for all panels in this figure.

-Figure 17: Note that the upper tail of O3 concentrations simulated by AM4 is over-predicted. Since this upper tail is important for characterizing O3 events above the NAAQS, this should be discussed in the enhanced model performance section recommended above.

-Figure 17 and S14: I suggest you add analogous figures that shows this same information but for CASTNET sites in the SW US at low elevation. It is important to be able to compare the results you are finding for high-elevation sites to what is predicted at lower elevation sites.

-Figure S6: Suggest showing the June 11-14 event in this figure as well.

-Table S1: suggest adding AM4 and GEOS-Chem model statistics of Apr-June 2016 O3 simulations to this table (mean bias, r etc).

-Table S2: Table states that MEGAN was used in AM4 simulations, but this is not clear in description in section 2.3. If MEGAN emissions were used for AM4, then add this to 1st paragraph of section 2.3. Also, Table refers to "section 2.4" which does not appear to exist in the paper.

-Note that the US O3 NAAQS is only exceeded when O3 is > than the level of the standard, not >= the level and that ozone values are truncated to the nearest ppb. Therefore, a measured concentration of 70.9 ppb is meeting the current NAAQS. When showing levels that violate the standard either in figures or tables, a line at 71 ppb would be more appropriate than a line at 70 ppb. For instance, in Figure S12, the cutoff for the blue bars should be >= to 71ppb.

References:

Dolwick, P, Akhtar, F, Baker, KR, Possiel, N, Simon, H and Tonnesen, G. 2015. Comparison of backÂňground ozone estimates over the western United States based on two separate model methodologies. Atmos Environ 109: 282–296. DOI: https://doi.org/10.1016/j.atmosenv.2015.01.005

Emery, C, Jung, J, Downey, N, Johnson, J, Jimenez, M, Yarwood, G and Morris, R. 2012. Regional and global modeling estimates of policy relevant backÂňground ozone over the United States. Atmos EnviÂňron 47: 206–217. DOI: https://doi.org/10.1016/j.atmosenv.2011.11.012

Henneman et al. 2018. Responses in Ozone and Its Production Efficiency Attributable to Recent and Future Emissions Changes in the Eastern United States. Environ. Sci. Technol., 51, 23, 13797-13805

Jaeglé, L., Shah, V., Thornton, J.A., Lopez-Hilfiker, F.D., Lee, B.H., McDuffie, E.E., Fibiger, D., Brown, S.S., Veres, P., Sparks, T.L., Ebben, C.J., Wooldridge, P.J., Kenagy, H.S., Cohen, R.C., Weinheimer, A.J., Campos, T.L., Montzka, D.D., Digangi, J.P., Wolfe, G.M., Hanisco, T., Schroder, J.C., Campuzano-Jost, P., Day, D.A., Jimenez, J.L., Sullivan, A.P., Guo, H., Weber, R.J., 2018. Nitrogen Oxides Emissions, Chemistry, Deposition, and Export Over the Northeast United States During the WINTER Aircraft Campaign. J. Geophys. Res. Atmos. 123, 12,368-12,393. doi:10.1029/2018JD029133

Jaffe, DA, et al. 2018. Scientific assessment of background ozone over the

U.S.: Implications for air quality management. Elem Sci Anth, 6: 56. DOI: https://doi.org/10.1525/elementa.309

Salmon, O.E., Shepson, P.B., Ren, X., He, H., Hall, D.L., Dickerson, R.R., Stirm, B.H., Brown, S.S., Fibiger, D.L., McDuffie, E.E., Campos, T.L., Gurney, K.R., Thornton, J.A., 2018. Top-Down Estimates of NO x and CO Emissions From Washington, D.C.-Baltimore During the WINTER Campaign. J. Geophys. Res. Atmos. doi:10.1029/2018JD028539
* * *

---

## Referee Comment (RC2) · Anonymous Referee #2 · 7 Feb 2020

This paper leverages intensive field measurements from the Fires, Asian, and Stratospheric Transport-Las Vegas Ozone Study (FAST-LVOS) and two global chemistry models to attribute high ozone events in the Las Vegas area to various causes: wildfires, stratosphere to troposphere ozone intrusions and transport from Asia. For many of the events the disagreement between the two models is notable.

Overall, I found this to be an interesting, well-written paper with well-crafted figures. After the authors address a few points I would recommend publication.

Major Comments:

I. It is unclear how the different events are attributed to different sources. A more clear

discussion of attribution is necessary.

(i) As the author's state: "Identifying the primary source of the high-O3 events solely based on observations is challenging; additional insights from models thus needed as we demonstrate below." Thus, how are the events attributed in Table 1? Is the measured data in section 4.1 used to support the modeled attribution? Or is it the primary attribution mechanism? Are both models used to attribute a particular event or just one? Is the event attribution through the preponderance of evidence?

(ii) On a more philosophical note it seems the authors often make good qualitative arguments for a particular type of event but is this sufficient to attribute an event to a particular cause? In particular, as air from various sources is mixes together how does one determine the cause of an ozone exceedance? Quantitatively, doesn't one need to know the average ozone contribution from a particular source and if ozone from that source is elevated sufficiently from its average one can attribute the ozone exceedance to that source (even if it is not the dominant source)? Or does one require a particular source to be the dominant source to attribute an event to it? At any rate more detail should be given as to what it means to attribute an event to a source.

II. The difference between the models in Figure 6 is deeply disturbing to me. While the authors concentrate on the differences in the stratospheric ozone intrusion, the figures overall are very different, not only in their ozone but in the isentropes. Is this a resolution problem, an interpolation problem, a meteorological analysis problem or possibly a result of differences in the advection algorithm? This seems quite important to determine as one of the main differences between the models seem to be in their handling of stratospheric intrusions. It seems as a minimum the authors could look at: (i) potential vorticity and potential temperature surfaces in the native resolution of the two meteorological datasets. Are these the same or different? (ii) Then examine potential vorticity and potential temperature in the resolution of the two model grids. Does changing the model grid do something to the fields? (iii) Finally they could examine the ozone differences between the figures. Is the ozone similar in the stratosphere in general in the two models which would point to greater numerical diffusion in GEOS-CHEM diluting the stratospheric intrusion? Or perhaps GEOS-CHEM has less ozone in the stratosphere. The paper seems to imply that the simplified stratospheric chemistry and dynamics in GEOS-CHEM is the reason for the discrepancy between the two models. What evidence do the authors have for this assertion?

III. When comparing differences between the AM4 and GEOS-CHEM model it would be useful to know the extent to which these differences might be due to differences in emissions. At a minimum the authors should discuss some of the emission differences, and the extent to which these might contribute to the difference in the USB ozone. Ideally these type of studies should be done with the same emissions. However, the authors should take any emission differences into account in their analysis, or at the least discuss that these may be a source of uncertainty in analyzing the differences between the models.

IV. One of the important aspects of this study seems to be in the attribution of exceedances. I think the authors should quantitatively assess the skill of the models in diagnosing exceedances (perhaps over a longer period of time than the FAST-LVOS timeframe itself, if possible). There are many measures of this skill in the literature. What percent of time when an exceedance is measured do each of the models predict the exceedance? And how often do the models get a false positive (predict an exceedance when one doesn't occur)?

V. I feel the conclusions could be made stronger. The authors give for the most part a detailed comparison between the two models. I think the paper would be strengthened if the authors stepped back a little.

(i) First, it would be interesting to discuss model skill in simulating the exceptional events as discussed in the first paragraph in the paper. Giving the difference between the two model simulations to what extent are we confident that they can screen exceptional events? What is the skill of the models in assessing extreme events (see

comment IV).

(ii) Second, in the case of an exceptional event, to what extent can these models be used to attribute the event to a particular cause?

(iii) The authors claim that much of the pattern of USB in AM4 is due to STT and the ability to simulate STT is an important difference between the models. This seems like an important conclusion, but how strong is the evidence? It would be good if the authors would summarize the reasons they conclude this. Do the differences in USB coincide with locations where the stratospheric tracer is high? Some more analysis might be beneficial to really make this point.

(iv) While the authors point out some differences in USB it would be nice to quantify the uncertainty here. I think a difference map in the USB between the two models would be very helpful. Do we know USB within 10% or 20% (at least based on these two models) and how important is this for policy considerations.

Minor Comments:

1. P2, l47 "contribute". Should this be contribute episodically?

2. P2, l58 "independent". This seems a bit strong. As has been shown in climate models (e.g., Knutti et al., 2013) models are really not independent from each other due to the sharing of information and algorithms across groups. For example the two models in this paper share the MEGAN scheme, but probably also share other aspects. Thus, I would delete the word "independent" here and in other locations.

3. Fig. 6, Please make the vertical and horizontal scales identical so these figures are easier to compare.

4. P6, l46 Please give the frequency of measurements used here and elsewhere.

5. P7, l188 What is GEOS-FP meteorology?

6. P8, l229 What is the standard representation of lightning?

7. P9, l235 Are the emissions the same in AM4 and its predecessor? If not to what extent is the comparison between them simply a matter of the different emissions. At the minimum the authors should mention these comparisons use different emissions and the extent to which these differences can explain the differences between the models.

8. P12, l319. PVU is not a unit. Please give the mks units for a PVU.

9. P14, l394. The authors suggest excessive lightning NOX in GEOS-chem causes excessive ozone. They cite a number of older papers to make their point. What is the evidence in this study for excessive lightning NOX? For example, P18 l496 the authors state the overestimate is likely due to lightning NOX. On P20 l569 the author categorically state it is the abundance of lightning NOx that results in higher background ozone in GEOS-chem. Without more analysis it seems lightning NOX is a possible explanation. However, if they authors claim this is the likely explanation they need to give some more evidence.

10. P14, section 4.3: The authors made a number of sensitivity simulations with respect to the simulation of fire plumes. I did not get a sense as to which of these sensitivities improved or degraded the simulation. Please give some overall conclusions.

11. P15, l14 "dominant source". Could you clarify? If the local emissions are 20-30 ppb and simulated emissions are over 60 ppb, why are local emissions the dominant source?

12. P17 l481 How were the STT events diagnosed?

13. P17 l486 "underestimates the magnitude of STT". The authors show that GEOS-CHEM underestimates the ozone concentrations in stratospheric folds, at least the ones they examined. First this sentence needs to be qualified as to where and when. Secondly while GEOS-CHEM may underestimate the ozone in the stratospheric intrusions this does not mean it underestimates STT (the model might simply be excessively

diffusive while still simulating the same exchange).

14. P18 l519, and p19 l20: "Many of the standard O3 events..." Could you quantify this? From the figure it appears to be less than half the events?

15. P19 l549 "contributing to ~30ppbv to surface ozone". I believe this is a modeling result. Please state this.

16. P20 l555 "wildfire event". This would be a good place to summarize whether any of the sensitivity tests resulted in a better capture of the wildfire event.

17. P18, l500 "likely reflect": what are the arguments for this? It might be interesting to show a difference map for USB, getting at the uncertainty in USB between two state-of-the-art models.

---

## Author Comment (AC1) · 4 May 2020

**Dear Dr. West:**

We truly appreciate the reviewers for carefully reading the manuscript and providing helpful suggestions. Below we include a point-by-point response (in bold blue) to the reviews, responding to their comments (in italic) and explaining the changes made to the manuscript (in light blue).

Best regards, Meiyun Lin (on behalf of the authors) April 25, 2020

**Reviewer* #1**

In this article, Zhang et al evaluate ambient and modeled data in the Las Vegas area during the 2017 FAST-LVOS study. They aim to determine dominant source categories for high observed ozone events. They leverage both enhanced monitoring data and sensitivity simulations from multiple global models to understand O3 events at this time and location. The evaluations shown here highlight the challenges with determining O3 sources. Even with these detailed datasets the evidence for categorizing O3 events on many of the event days is not definitive. I think this analysis is a valuable addition to the literature, but I think the uncertainties in the analysis need to be more clearly communicated. I believe that the authors have overstated the confidence in their ability to categorize ozone events from the data provided. In addition, there are certain areas where the article needs additional details and background, specifically: 1) include references to additional relevant articles on background ozone, 2) include more comprehensive model evaluation information especially as it relates to the specific times and locations of the ozone events of interest, 3) provide more systematic information with which to compare indicator values in each these events through Table 1 or additional tables/figures.

**RE:** Thank you for the comments. We have revised the manuscript following your suggestions.**

Major comments:

2. Introduction: Please reference the recent comprehensive review of background ozone by Jaffe et al (2018). For the paragraph summarizing past modeling to predict USB, please also include references to Dolwick et al (2015), Emery et al (2012) and additional references cited in Jaffe et al (2018). In lines 80-85, please note that the Dolwick et al (2015) modeling included analysis of daily O3 and plotted USB against total O3 showing the range of daily values. Jaffe et al (2018) also included daily quantification of background ozone. Similarly, when discussing past literature on USB estimates in section 5, please also cite and compare to Dolwick et al (2015) and Emery et al (2012).

**RE:** The relevant discussion in the Introduction has been rephrased (see Lines 80-85):**

"Large inter-model differences not only exist in seasonal means but also in day-to-day variability (e.g., Fiore et al., 2014; Dolwick et al., 2015; Jaffe et al., 2018). An event-oriented multi-model comparison,

tied closely to intensive field measurements, is needed to provide process insights into the model discrepancy."

We have also cited Dolwick et al (2015) and Emery et al (2012) in Section 5 (see Lines 1040-1050).

**2. Modeling description and evaluation:**

- Given the heavy reliance on AM4 results to categorize events in this paper, the authors should provide more detailed information on model performance. Model performance is only shown for aloft measurements. I suggest adding model performance of ground-level O3 based on measurements at CASTNET sites. Also, it would be useful to report mean bias of ground-level O3 (based on nearby CASTNET and FAST-LVOS measurement) for each specific episode day (i.e. what is the model performance at the times and locations of interest and how does it change on different days examined). This will provide important context for interpreting modeling results used to classify the different O3 episodes. This could be done as part of Table 1 or as a separate table in the paper. In addition, performance information that is already available in Figures 15 and 17 could be brought forward and expanded upon in section 3.

**RE:** We have moved the time series analysis of MDA8 O3 to Section 3 and added the statistics for model performance at all CASTNet sites in Table S1. The purpose of Section 3 is to provide an overall evaluation of the models. The event-specific evaluation of the models is discussed in Section 4.

-Given 50 km resolution of AM4, the model may be better suited to quantify O3 from some sources than from others. The coarse resolution may not matter as much for stratospheric intrusions and for transport from Asia but may be insufficient for capturing photochemical production from sources that may have gradients in precursor emissions (fires and local/regional US anthropogenic sources).

**RE:** We agree with the reviewer that the 50-km resolution of AM4 may be insufficient to represent the local-scale photochemical production. We have added some related discussions in Section 4.4 – Regional and local anthropogenic pollution events.

-The AM4 model simulations were conducted for January-June 2017 and the first episode evaluated was in late April. That corresponds to 4-month spin-up period which seems short for global simulations that are tracking impact from long-range sources. Please address this short spin-up period? Why is this length of spin-up appropriate for the simulations conducted here?

**RE:** This is now clarified in the revised manuscript (Line 197):

"The high-resolution BASE and sensitivity simulations for January – June 2017 are initialized from the corresponding nudged C96 ( $\sim$ 100x100 km2) simulations spanning from 2009 to 2016 (8 years)."

-A stratospheric tracer is implemented in AM4. Please provide details on whether this tracer is inert or reactive (i.e. can be degraded by chemistry and deposition). If the tracer is inert, then it should not be used to quantify stratospheric impacts (e.g. line 313) because the lack of degredation processes will lead to an overestimate of stratospheric O3 influence. In this case it could still be used qualitatively to identify times and locations of stratospheric influence.

**RE:** Thank you for the comment. This is clarified in the revised manuscript (Lines 178-182; Section 2.2).**

"We implement a stratospheric  $O_3$  tracer (O3Strat) in GFDL-AM4 to track O3 originating from the stratosphere. The O3Strat is defined relative to a dynamically varying e90 tropopause (Prather et al., 2011) and is subject to chemical loss in the same manner as odd oxygen of tropospheric origin and deposition to the surface (Lin et al., 2012a; Lin et al., 2015a)."

-GEOS-Chem description did not specify the simulation period. Was this also Jan-June 2017? If so, please also address the relatively short spin-up on the GEOS-Chem simulations.

**RE:** The GEOS-Chem simulations period and the spin-up process are now clarified in the revised manuscript (Line 223).**

"We conduct two nested high-resolution simulations with GEOS-Chem for February-June 2017: BASE and a USB simulation with anthropogenic emissions zeroed out in the U.S. (Table S3). Initial and boundary conditions for chemical fields in the nested-grid simulations were provided by the corresponding BASE and USB GEOS-Chem global simulations at  $2^{\circ} \times 2.5^{\circ}$  resolution for January-June 2017. Only simulations during April-June are analysed in this study. The threemonth spin-up period (January-March) used for GEOS-Chem is relatively short compared to the multi-year GFDL-AM4 simulations, although it should be sufficient given that the lifetime of ozone in the free troposphere is approximately three weeks (e.g., Young et al., 2016)."

-I have several questions/comments on the US emissions adjustment. You state that NOx emissions were cut by 50% in the Eastern US, does this mean the NOx emissions were left unchanged in the Wester US? The Travis et al paper was based on an analysis for 2013 and NOx emissions and vehicle fleet characteristics (age, vehicle emissions control systems etc) have been continually changing in the US. Is it appropriate to apply scaling factors based on a 2013 analysis to this 2017 time period? In addition, GEOS-Chem emissions already account for decreasing NOx emissions in more recent years based on EPA trends information. Since the EPA trends include improvements to inputs for mobile source emissions calculations in more recent years, this 50% adjustment may be double-correcting for adjustments that are already included in the more recent EPA data. In addition, recent papers coming out of the 2017 WINTER campaign (Salmon et al., 2018; Jaegle et al. 2018) suggested that EPA NOx emissions were unbiased in winter, so should same 50% NOx cut be applied in winter months as in summer? If this adjustment was only made in the Eastern US, perhaps the impact of these adjustment are limited for this analysis which focuses on the Las Vegas area. Also, why do you use monthly climatology for lightning NO rather

than a method based on NLDN? Perhaps the impact of lightning NO representation on model performance noted on Line 387 would be less if actual emissions rather than monthly climatology were used. It would be useful to add a table to the supplemental information that included emissions levels (tons of NOx, CO, VOC) by region (Eastern US, Western US, China, EU, Fires etc) used for the 2 models.

**RE:** We only apply the NOx emissions adjustments to the Eastern US. To our knowledge, the findings of Travis et al. are not just limited to the NEI 2013 used in their study. They imply that NOx emissions over the eastern U.S. in all global emission inventories may be overestimated since almost all of the current global models using a varies of emission inventories in different years overestimate surface ozone in the southeast US.

Comparison of regional NOx, CO, and NMVOCs emissions from AM4 and GC is shown in Table S1.

3. Source characterization for specific O3 events:

For many of the episodes listed in table 1, the evidence is suggestive but not compelling for the classifications given. I suggest adding to Table 1 a characterization of how confident you are in the classification. Based on evidence presented in the paper, the only episode that I would rate a "high confidence" is the June 11th stratospheric intrusion. I suggest adding fields to Table 1 so that episodes can be more easily compared: H2O mixing ratio (Avg +/- SD), O3:CO slope, O3/NOz. Several of these indicators are mentioned for one of the episode days in the text but not provided for all days so that the reader can compare what they look like during different types of events. It would be useful if Table 1 provided a more detailed accounting of which lines of evidence were used to classify each of the episodes examined. It seems that in most cases, the ambient data can be indicative of influence from different types of sources but can't provide quantitative estimates of how much O3 is from US sources versus fires/stratosphere/Asia etc. The model zero out simulations are often the basis for determining that O3 during an episode is primarily from one source. Given this dependence on the model, more detail on model methods and model performance compared to ground-based measurement on each of the episode days is warranted.

**RE:** We have added $H_2O$ mixing ratios in Table 1 and revised the event classification to reflect uncertainties. We discussed in the text how the attribution in Table 1 is done (Lines 325-400):**

"The attribution is based on a combination of observational and modeling analyses. First, we examine the O3/CO/H2O relationships and collocated meteorological measurements from the NOAA/ESRL mobile lab deployed at Angel Peak to provide a first guess on the possible sources of the observed high-O3 events (Section 4.1). Then, we analyze large-scale meteorological fields (e.g., potential vorticity), satellite images (e.g., AIRS CO), and lidar and ozonesonde observations to examine if the transport patterns, the high-O3 layers and related tracers are consistent with the key characteristics of a particular source (Section 4.2-4.5). Available aerosol backscatter measurements and multi-tracer aircraft profiles are also used to support the attribution (Sections 4.3 and 4.6). Finally, for each event we examine the spatiotemporal correlations

of model simulations of total  $O_3$ , background  $O_3$ , and its components (e.g., stratospheric ozone tracer), both in the free troposphere and at the surface. For a source to be classified as the dominant driver of an event,  $O_3$  from that source must be elevated sufficiently from its mean baseline value. ".

**3.1. Even characterization for stratospheric intrusions:**

-Apr 22-23 and May 13-14 - no details are provided to evaluate these events

**RE:** These two events occurred before the FAST-LVOS campaign began. However, the stratospheric influence on these two events is evident in the time series analysis of MDA8 O3 now shown in Fig.4, as we discussed in Section 3.2. More detailed analyses are provided in the Supplemental Material. We have clarified these in Table 1.

-June 11-13 - anti-correlated O3:CO and very dry conditions are compelling for predominant stratospheric impact. I suggest adding this episode to figures S5 and S6 to show what the stratospheric tracer looks like for these days.

**RE:** Thanks. We have added anomalies in AM4 stratospheric ozone tracer for June 11-14 in Fig.9 and discussed this in the main article (Section 4.2, Lines 530-535).**

"Over the areas where observed MDA8 O3 levels are 60–75 ppbv, GFDL-AM4 estimates 50–65 ppbv USB O3 with simulated O3Strat 20–40 ppbv higher than its mean baseline level in June."

-June 14 - Evidence not compelling. O3:CO is positively correlated and slope is similar to slope from "US anthropogenic events". O3/NOz evidence not provided for any other events for comparison and reference from 1998 analysis may not be relevant to current conditions. Note that a recent paper by Henneman et al (2017) found that O3/NOz values have increased over time as US NOx levels have decrease leading to more efficient O3 production. They reported that for Atlanta, the O3/NOz increased from 5.4 in 2001 to 9.3 in 2011. Similarly, in urban Gulfport, FL the ratio increased from 11.1 in 2001 to 20.5 in 2011. So, the range of 1-7 based on Kleinman et al (2002) may not be relevant for current urban conditions in the US. H2O mixing ratio and O3:CO slope appear to be in same range as other events classified as from US anthropogenic influence. Vertical profiles and modeling impacts are not compelling; vertical profiles look similar to what I would expect for local formation with low overnight PBL concentrations due to titration, some mixing down of residual layer in the morning accompanied by increasing boundary layer O3 during the day. What does the stratospheric tracer look like on this day?

**RE:** The June 14 event likely resulted from mixing of regional anthropogenic pollution and transported stratospheric O3 residual from the previous day, as we clarified in Table 1 and discussed in the revised manuscript:

"Mixing of stratospheric ozone with regional pollution on June 14:

Stratospheric air masses that penetrate deep into the troposphere can mix with regional anthropogenic pollution and gradually lose their typical stratospheric characteristics (cold and dry air containing low levels of CO), challenging diagnosis of stratospheric impacts based directly on observations (Cooper et al., 2004; Lin et al., 2012b; Trickl et al., 2016). On June 14, O3 measured at Angel Peak is positively correlated with CO ( $\Delta O_3/\Delta CO = 0.75$ ; Fig. 6b), similar to conditions of anthropogenic pollution on June 16 (Fig. 6c–d). TOPAZ lidar shows elevated O3 of 70–80 ppbv concentrated within the boundary layer below 3 km altitude (Fig. 8b). These observational data do not provide compelling evidence for stratospheric influence. However, GFDL-AM4 simulates elevated O3Strat coinciding with the observed and modeled total O3 enhancements within the PBL, indicating that O3 from the deep stratospheric intrusion on the previous day may have been mixed with regional anthropogenic pollution to elevate O3 in the PBL. At the surface (the bottom panel in Fig.9), AM4 simulates high USB O3 and elevated O3Strat (20-40 ppb above its mean baseline) over Arizona and New Mexico where MDA8 O3 greater than 70 ppb were observed. The fact that GEOS-Chem is unable to simulate the ozone enhancements in lidar measurements and at the surface further supports the possible stratospheric influence. This case study demonstrates the value of integrating observational and modeling analysis for the attribution of high-O3 events over a region with complex O3 sources."

**We agree that the discussion on O3/NOz ratios is not relevant here and thus have deleted it from the revised manuscript.**

**3.2.** Event characterization for wildfires (June 22): The increased aerosol backscatter aloft on the previous day (Figure 14) is suggestive that wildfires may be advected to this general location. The O3:CO and H2O mixing ratio is not convincing. Why are O3 and CO not correlated if they are both originating from the fire event? Also, the H2O mixing ratio looks like it is in the range of what was observed on June 16th, a "US anthropogenic emissions" event. What are typical H2O mixing ratios (average +/- SD) for Las Vegas during June? Is 3.5 g/kg really outside what you might get from normal meteorological variation? In fig 9 it is clear that the models cannot accurately capture O3 on this day, so it does not appear that modeling evidence should be used here leaving us with no way to quantify the impact of wildfires on the ground-level ozone. Do the fire sensitivities described on lines 394-406 impact the vertical profiles of O3? If you were to recreate Figure 9 using those sensitivities would the models be more able to capture O3 vertical profile?

**RE:** We agree with the reviewer that the increased aerosol backscatter aloft is a more compelling evidence of the wildfire influence, in contrast to the O3/CO/H2O relationship, during this event. Thus, we have moved the aerosol backscatter analysis (previously Fig.14) to Section 4.3 (now Fig.10) to support the discussions. Indeed, the models have difficulty accurately simulating the observed ozone enhancements during this event. We have stated

**the uncertainties both in Section 4.3 and in the Conclusions as the reviewer suggested. Particularly, in the Conclusions, we stated:**

"The two models also differ substantially in total and background  $O_3$  simulations during the June 22 wildfire event. GEOS-Chem captures the broad  $O_3$  enhancement in lidar observations but overestimates surface MDA8  $O_3$  at some sites during this event. It remains unclear whether higher USB  $O_3$  simulated by GEOS-Chem during this event is from greater  $O_3$  produced from wildfire emissions or excessive lightning NOx emissions in the model. Although GFDL-AM3 captures the observed interannual variability in  $O_3$  enhancements from large-scale wildfires over the WUS (Lin et al., 2017), GFDL-AM4 has difficulty simulating the observed  $O_3$  enhancements during the relatively small-scale wildfire event on June 22. Sensitivity simulations with fire emissions constrained at the surface or with part of fire NOx emissions emitted as PAN do not substantially improve simulated  $O_3$  on June 22. Wildfires typically occur under hot and dry conditions which also enable the buildup of  $O_3$  produced from regional anthropogenic emissions, complicating an unambiguous attribution of the high- $O_3$  events solely based on observations. Screening of exceptional events due to wildfire emissions remains a serious challenge."

**Regarding the poor correlation between CO and O3 measured at Angel Peak, we added the following discussions in Section 4.1:**

"In particular, exceptionally high CO levels (~100–440 ppbv) on June 22 (Fig. 6e) suggest influences from wildfires. Ozone enhancements were measured by the TOPAZ ozone lidar on June 22 (Section 4.3) although the correlation between CO and O3 at Angel Peak is not strong. The net production of O3 by wildfires is highly variable, with many contradictory observations reported in the literature (Jaffe and Wigder 2012). The amount of O3 within a given smoke plume varies with distance from the fire and depends on the plume injection height, smoke density, and cloud cover (Faloona et al., 2020)."

**3.3. Event characterization for regional/local pollution events:**

-June 16th: O3 and CO are positively correlated suggesting that O3 formed from an emissions source that also emitted CO. This appears to be based primarily on modeled predictions of total O3 and USB. How does this compare to other events? What is the mean bias for ground-level O3 in Las Vegas on this day? Please state the level of elevated ozone aloft measured by TOPAZ (55-70 ppb?). June 2 and June 29-30: Not much information provided.

**RE:** We have revised discussions on the regional pollution events: June 16, June 2, and June 29-30. Please see tracked changes in Section 4.4.**

Event characterization for Asian transport event (May 24): AIRS CO images from the days leading up to May 24 are suggestive of transport from Asia. No information is provided on O3: CO or H2O mixing ratio. Modeling from AM4 predicts 5-10 ppb influence from Asia. Is 5-10 ppb of O3 from Asia enough to classify this as "Asian transport" alone since O3 as > 70 ppb? Doesn't this suggest that there was at least one other major source? What is the MB for O3 at ground-level in Las Vegas on this day? Also note that the elevated O3 at 1-4 km altitude could also be local/regional O3 in the residual layer from previous days. The elevated O3 at 6-8 km is more definitively long-range transport. For this episode, I would say it is inconclusive. The models predict an Asian contribution, but the observations are not conclusive as they could also be showing local/regional photochemical production over several days. The evidence from AIRS is suggestive but cannot provide any quantification of how much of the O3 comes from Asia.

**RE:** We agree completely that elevated ozone at 6-8 km altitude reflects the Asian influence while elevated O3 at 1-4 km altitude appears to be influenced by local/regional pollution. We have revised the discussions accordingly. Particularly, we discuss whether the Asian contribution is sufficiently larger than its average value, as suggested by Reviewer #2:

"Both GFDL-AM4 and GEOS-Chem capture the observed  $O_3$ -rich plumes at surface–4 km and 6–8 km altitude above Clark County during this event. Elevated  $O_3$  at 6-8 km altitude reflects the long-range transport from Asia, as supported by concurrent enhancements in total and USB  $O_3$  in both models and by the large difference in the AM4 BASE simulation and the sensitivity simulation with Asian anthropogenic emissions zeroed out. Elevated  $O_3$  at 1-4 km altitude appeared to be influenced by a residual pollution layer from the previous day; this plume was later mixed into the growing PBL (up to 4 km altitude), elevating MDA8  $O_3$  in surface air on May 24. Further supporting the impact from regional or local pollution below 4 km altitude, both models simulate much larger enhancements in total  $O_3$  (70-90 ppbv) than in USB  $O_3$  (~50 ppbv).

On May 24th, MDA8 O3 approached or exceeded the 70-ppbv NAAQS at multiple sites in California, Idaho, Wyoming, and Nevada (Fig. 15a), likely reflecting the combined influence from regional pollution and the long-range transport of Asian pollution. MDA8 O3 at four surface sites in Clark County was above 65 ppbv. More exceedances would have occurred if the level for the NAAQS were lowered to 65 ppbv. In parts of Idaho, Wyoming, California where observed MDA8 O3 were higher than 60 ppbv, the contribution of Asian anthropogenic emissions as estimated by GFDL-AM4 were 8–15 ppbv (Fig. 15a), much higher than the springtime average contribution of ~5 pppv estimated by previous studies (e.g., Lin et al., 2012), supporting the episodic influence from Asian pollution during this event. At several high-elevation sites in California such as Arden Peak (72 ppbv) and Yosemite National Park (70 ppbv) where observed MDA8 O3 exceeds the NAAQS level, the contribution of Asian pollution is approximately 9 ppbv. Ozone produced from regional and local anthropogenic emissions dominates the observed MDA8 O3 above 70 ppbv in the Central Valley of California. "

3.4. Event characterization for unattributed event (June 28): Much of the evidence here seems to rule out rather than support any particular source. The elevated filament seems uncharacteristic of longer range transport over which time the filament is likely to disperse. Yet the satellite CO suggests possible transport form Asia. The back trajectories suggest possible influence from a California wildfire but the relatively dry air and lack of elevated aerosol backscatter makes it seem unlikely that a narrow filament of O3 could be from a wildfire without substantially elevated PM. The narrow filament of O3 could be from the stratosphere but the elevated CO and CH4 suggest photochemical formation. It should also be noted that Sequoia NP often receives pollution from California's San Joaquin Valley and Los Angeles, so the back trajectories make it seem equally likely that the source is US emissions vs a wildfire. How do the deltaO3/deltaCO and H2O mixing ratio compare to other events. Is the O3:CO slope statistically different from slopes on other days (using 95 % CO from the regression)? Could you use a T-test on the mixing ratio to determine whether it is statistically different from H2O levels on other episode days? One line 307, you indicate that the dryer air during this episode indicates the plume was from Asia or Los Angeles, which is a different conclusion than you present on lines 470-472 that states it is likely from a wildfire plume. What is the model mean bias (in ppb) for ground-level O3 on this day?

**RE:** Indeed, discussions for this event rely heavily on limited observational evidence. We have revised the discussions and provided information you asked for when possible. Please see tracked changes in Section 4.6.

**4.** Conclusions: I think the language in the conclusion is too strong since there appears to be a lot of uncertainty in the characterization of O3 sources for most of the events evaluated in this paper. For instance, I do not believe that this analysis has been successful in "pinpointing sources of observed MDA8 O3" (line 536) and the language in general should reflect the uncertain nature of the conclusions that can be drawn for the analysis so far.

**RE:** Thank you for the suggestion. We have revised the Conclusions to reflect uncertainties. Please see tracked changes in Section 6. Specifically, we follow the suggestions by Reviewer #2 and discuss whether the models are useful for screening of exceptional events due to STT and wildfires.

Other specific comments:

1. Line 300: did you mean to indicate specific panels of Figure 5?

**RE: Revised.**

2. Figure 6: suggest marking location of O3 sonde in panel a)

**RE: Done.**

3. Figure 7: Suggest using a separate color palette for the rightmost panel of AM4 rather than scaling by 2.5 to avoid confusion by the reader.

**RE:** Considering that Fig 7 is large and busy, a separate color palette will make the figure difficult to read. We've stated in the figure caption that the stratospheric ozone tracer was scaled by 2.5.**

**4.** Figure 15: Some but not all of these monitors are shown in the map in Figure 1. I suggest you provide a map that has locations of all monitors used in Figure 15.

**RE:** Revised. Now all sites are shown in Figure 1 and Table S1.

**5.** Figure 16: Why do you use 2 different scales in panels a) and b)? It looks like the O3 covers the same range of values in both. It is confusing to the reader when trying to compare the results from the various panels. I suggest just using the color range from panel a) for all panels in this figure.

**RE: Done.**

**6.** Figure 17: Note that the upper tail of O3 concentrations simulated by AM4 is overpredicted. Since this upper tail is important for characterizing O3 events above the NAAQS, this should be discussed in the enhanced model performance section recommended above.

**RE:** Thank you. We have added the discussions in the revised manuscript.

**7.** Figure 17 and S14: I suggest you add analogous figures that shows this same information but for CASTNET sites in the SW US at low elevation. It is important to be able to compare the results you are finding for high-elevation sites to what is predicted at lower elevation sites.

**RE:** Good suggestion. There are only a few low-elevation CASTNET sites in the SWUS. We aid some analyses for the AQS sites in the Clark County. Due to coarse model resolution (particularly AM4), most of the sites reside in the same model grid, making the comparison not meaningful. We agree completely with the reviewer that it is important to show the differences between high-elevation and low-elevation sites. We plan to do this in a forthcoming study using a higher resolution version of the models.

**8.** Figure S6: Suggest showing the June 11-14 event in this figure as well.

**RE:** Anomalies in AM4 O3Strat for the June 11-14 event are now shown in Fig.9 in the main article.

**9.** Table S1: suggest adding AM4 and GEOS-Chem model statistics of Apr-June 2016 O3 simulations to this table (mean bias, r etc).

**RE:** Done and the results are briefly discussed in Section 3.

**10.** Table S2: Table states that MEGAN was used in AM4 simulations, but this is not clear in description in section 2.3. If MEGAN emissions were used for AM4, then add this to 1st paragraph of section 2.3. Also, Table refers to "section 2.4" which does not appear to exist in the paper.

**RE:** This is now clarified in Section 2.3.**

11. Note that the US O3 NAAQS is only exceeded when O3 is > than the level of the standard, not >= the level and that ozone values are truncated to the nearest ppb. Therefore, a measured concentration of 70.9 ppb is meeting the current NAAQS. When showing levels that violate the standard either in figures or tables, a line at 71 ppb would be more appropriate than a line at 70 ppb. For instance, in Figure S12, the cutoff for the blue bars should be >= to 71ppb.

RE: Thank you for your insight. We have avoided using the term such as "standard-exceeding" as it has a specific policy meaning.

**Response to Reviewer #2**

**Reviewer* #2**

This paper leverages intensive field measurements from the Fires, Asian, and Stratospheric Transport-Las Vegas Ozone Study (FAST-LVOS) and two global chemistry models to attribute high ozone events in the Las Vegas area to various causes: wildfires, stratosphere to troposphere ozone intrusions and transport from Asia. For many of the events the disagreement between the two models is notable. Overall, I found this to be an interesting, well-written paper with well-crafted figures. After the authors address a few points I would recommend publication.

**RE:** Thank you for your supporting comments. We have revised the manuscript following your suggestions.

**Major Comments:**

*I.* It is unclear how the different events are attributed to different sources. A more clear discussion of attribution is necessary.

(i) As the author's state: "Identifying the primary source of the high-O3 events solely based on observations is challenging; additional insights from models thus needed as we demonstrate below." Thus, how are the events attributed in Table 1? Is the measured data in section 4.1 used to support the modeled attribution? Or is it the primary attribution mechanism? Are both models used to attribute a particular event or just one? Is the event attribution through the preponderance of evidence?

(ii) On a more philosophical note it seems the authors often make good qualitative arguments for a particular type of event but is this sufficient to attribute an event to a particular cause? In particular, as air from various sources is mixes together how does one determine the cause of an ozone exceedance? Quantitatively, doesn't one need to know the average ozone contribution from a particular source and if ozone from that source is elevated sufficiently from its average one can attribute the ozone exceedance to that source (even if it is not the dominant source)? Or does one require a particular source to be the dominant source to attribute an event to it? At any rate more detail should be given as to what it means to attribute an event to a source.

**RE:** Thank you for the suggestion. In Lines 325-400, we have clarified how the attribution in Table 1 is done. Please see also our Reponses to Reviewer 2 - Source characterization for specific O3 events. For a source to be classified as the dominant driver of an event, O3 from that source must be elevated sufficiently from its mean baseline value. We have followed this guideline throughout the manuscript for discussions of each event. For example, for the STT events, we show that on days and locations when observed MDA8 O3 exceeded 65 ppbv, AM4 O3Strat is 20-40 ppbv above its mean baseline level.

**II.** The difference between the models in Figure 6 is deeply disturbing to me. While the authors concentrate on the differences in the stratospheric ozone intrusion, the figures overall are very different, not only in their ozone but in the isentropes. Is this a resolution problem, an interpolation problem, a meteorological analysis problem or possibly a result of differences in the advection algorithm? This seems quite important to determine as one of the main differences between the models seem to be in their handling of stratospheric intrusions. It seems as a minimum the authors could look at: (i) potential vorticity and potential temperature surfaces in the native resolution of the two meteorological datasets. Are these the same or different? (ii) Then examine potential vorticity and potential temperature in the resolution of the two model grids. Does changing the model grid do something to the fields? (iii) Finally they could examine the ozone differences between the figures. Is the ozone similar in the stratosphere in general in the two models which would point to greater numerical diffusion in GEOS-CHEM diluting the stratospheric intrusion? Or perhaps GEOS-CHEM has less ozone in the stratosphere. The paper seems to imply that the simplified stratospheric chemistry and dynamics in GEOS-CHEM is the reason for the discrepancy between the two models. What evidence do the authors have for this assertion?

**RE:** Thank you for your insight. We have revised Fig.6 and the isentropes between the two models are similar. We have also conducted the additional analysis suggested by the reviewers. As shown in Supplemental Fig.S6, the synoptic-scale patterns of potential vorticity at 250 hPa and ozone levels in the UTLS are similar between the two models. We added the following statement in the revised manuscript:

"For comparison, GEOS-Chem simulates a much weaker and shallower intrusion (Fig. 7b), despite the similar synoptic-scale patterns of potential vorticity at 250 hPa and ozone levels in the UTLS (Fig.S6), suggesting possibly greater numerical diffusion in GEOS-CHEM diluting the stratospheric intrusion"

**III.** When comparing differences between the AM4 and GEOS-CHEM model it would be useful to know the extent to which these differences might be due to differences in emissions. At a minimum

the authors should discuss some of the emission differences, and the extent to which these might contribute to the difference in the USB ozone. Ideally these type of studies should be done with the same emissions. However, the authors should take any emission differences into account in their analysis, or at the least discuss that these may be a source of uncertainty in analyzing the differences between the models.

**RE:** Good point! We have summarized regional anthropogenic and fire emissions of NOx, CO, and NMVOCs in the U.S, China, and Europe in Table 1. Overall, the differences in anthropogenic and fire emissions used in the models are very small. The largest emission discrepancies are likely from lightening NOx emissions as we discussed in the paper. Unfortunately, lightening NOx calculated by the models are not archived in these simulations.

**IV.** One of the important aspects of this study seems to be in the attribution of exceedances. I think the authors should quantitatively assess the skill of the models in diagnosing exceedances (perhaps over a longer period of time than the FAST-LVOS timeframe itself, if possible). There are many measures of this skill in the literature. What percent of time when an exceedance is measured do each of the models predict the exceedance? And how often do the models get a false positive (predict an exceedance when one doesn't occur)?

**RE:** We now report the percentage of site-days with MDA8 O3 above 70 or 65 ppbv at Clark County and CASTNet sites in Table S5. The results are briefly discussed in Section 5.**

" Tables S4 and S5 report year-to-year variability in the percentage of site-days with springtime MDA8  $O_3$  above 70 ppbv (or 65 ppbv) and simulated USB levels during 2010–2017. The percentage of site-days with MDA8  $O_3$  above 70 ppbv during April–June 2017 is 0.9% from observations at CASTNet sites, 2.0% from GFDL-AM4, and 0.1% from GEOS-Chem. GFDL-AM4 captures some aspects of the observed year-to-year variability despite mean-state biases. For example, the observed percentage of site-days with MDA8  $O_3$  above 70 ppbv at CASTNet sites is highest (9.4%) in April-June 2012, compared to 3.1±3.2% for the 2010–2017 average. That statistics from GFDL-AM4 are 7.7% for 2012 and 4.0±2.9% for the 2010–2017 average. May–June mean USB MDA8  $O_3$  at Clark County sites are 50.9 ppbv in 2017, 55.3 ppbv in 2012, and 52.3±2.0 ppbv for the 2010–2017 average. Supporting the conclusions of Lin et al. (2015a), these results indicate that background  $O_3$ , particularly the stratospheric influence, is an important source of the observed year-to-year variability in high- $O_3$  events over the WUS during spring."

*V.* I feel the conclusions could be made stronger. The authors give for the most part a detailed comparison between the two models. I think the paper would be strengthened if the authors stepped back a little.

(i) First, it would be interesting to discuss model skill in simulating the exceptional events as discussed in the first paragraph in the paper. Giving the difference between the two model

simulations to what extent are we confident that they can screen exceptional events? What is the skill of the models in assessing extreme events (see comment IV).

(*ii*) Second, in the case of an exceptional event, to what extent can these models be used to attribute the event to a particular cause?

(iii) The authors claim that much of the pattern of USB in AM4 is due to STT and the ability to simulate STT is an important difference between the models. This seems like an important conclusion, but how strong is the evidence? It would be good if the authors would summarize the reasons they conclude this. Do the differences in USB coincide with locations where the stratospheric tracer is high? Some more analysis might be beneficial to really make this point.

(iv) While the authors point out some differences in USB it would be nice to quantify the uncertainty here. I think a difference map in the USB between the two models would be very helpful. Do we know USB within 10% or 20% (at least based on these two models) and how important is this for policy considerations?

**RE:** Thank you for the suggestion. We have revised the Conclusions substantially to include discussions on uncertainties and whether the models can be used for screening of exceptional events due to STT or wildfires. Please see tracked changes in Section 6.

Minor Comments:

*1. P2*, *l47* "contribute". Should this be contribute episodically?

**RE: Yes. Revised.**

**2.** P2, 158 "independent". This seems a bit strong. As has been shown in climate models (e.g., Knutti et al., 2013) models are really not independent from each other due to the sharing of information and algorithms across groups. For example the two models in this paper share the MEGAN scheme, but probably also share other aspects. Thus, I would delete the word "independent" here and in other locations.

**RE:** Good point. We have deleted "independent" in the revised manuscript.

3. Fig. 6, Please make the vertical and horizontal scales identical so these figures are easier to compare.

**RE: Revised as suggested.**

4. P6, 146 Please give the frequency of measurements used here and elsewhere.

**RE: Done.**

**5. P7, 1188 What is GEOS-FP meteorology?**

**RE:** Clarified in the revised manuscript:**

"... using the Goddard Earth Observing System – Forward Processing (GEOS-FP) assimilated meteorological data."

**6. P8, 1229 What is the standard representation of lightning?**

**RE: Rephrased to:**

"The model calculates lightning  $NO_X$  emissions using monthly climatology of satellite lightning observations coupled to model deep convection (Murray et al., 2012). The calculation of lightning  $NO_X$  in this study differs from that in Zhang et al. (2014), who used the U.S. National Lightning Detection Network (NLDN) data to constrain model flash rates."

7. P9, l235 Are the emissions the same in AM4 and its predecessor? If not to what extent is the comparison between them simply a matter of the different emissions. At the minimum the authors should mention these comparisons use different emissions and the extent to which these differences can explain the differences between the models.

**RE:** We noted in the revised manuscript:**

"Since the largest differences between the two models occur during the cold months, the difference in emissions inventories is unlikely the major cause."

8. P12, 1319. PVU is not a unit. Please give the mks units for a PVU.

**RE: Done.**

**9.** P14, 1394. The authors suggest excessive lightning NOX in GEOS-chem causes excessive ozone. They cite a number of older papers to make their point. What is the evidence in this study for excessive lightning NOX? For example, P18 1496 the authors state the overestimate is likely due to lightning NOX. On P20 1569 the author categorically state it is the abundance of lightning NOX that results in higher background ozone in GEOS-chem. Without more analysis it seems lightning NOX is a possible explanation. However, if they authors claim this is the likely explanation they need to give some more evidence.

**RE:** The influence of excessive lightning NOx emissions on the overestimates of background ozone has been discussed in detail by Zhang et al. (ACP, 2014). We have cited Zhang et al. (ACP, 2014) to support our discussions.

**10.** P14, section 4.3: The authors made a number of sensitivity simulations with respect to the simulation of fire plumes. I did not get a sense as to which of these sensitivities improved or degraded the simulation. Please give some overall conclusions.

**RE: Done. Please see Line 783 and the Conclusion section.**

**11.** *P15, l14 "dominant source". Could you clarify? If the local emissions are 20-30 ppb and simulated emissions are over 60 ppb, why are local emissions the dominant source?*

**RE: Rephrased to**

"Both models show boundary layer O3 enhancements in total O3 simulations but not in USB simulations (Fig. 11b), indicating that regional or local anthropogenic emissions are the primary source of observed O3 enhancements."

12. P17 1481 How were the STT events diagnosed?

**RE:** That sentence has been rephrased and moved to Section 3. We now say:

"The two models show large differences in simulated total and USB O3 on days when AM4 O3Strat indicates a stratospheric influence (highlighted in blue shading). AM4 O3Strat indicates frequent STT events during April–June with MDA8 O3 exceeding or approaching the current NAAQS of 70 ppbv."

**13.** P17 1486 "underestimates the magnitude of STT". The authors show that GEOSCHEM underestimates the ozone concentrations in stratospheric folds, at least the ones they examined. First this sentence needs to be qualified as to where and when. Secondly while GEOS-CHEM may underestimate the ozone in the stratospheric intrusions this does not mean it underestimates STT (the model might simply be excessively diffusive while still simulating the same exchange).

**RE:** That sentence has been rephrased and moved to Section 3.2. Lines 310-320:

"The two models show large differences in simulated total and USB O3 on days when AM4 O3Strat indicates a stratospheric influence (highlighted in blue shading)."

**14.** P18 l519, and p19 l20: "Many of the standard O3 events. . ." Could you quantify this? From the figure it appears to be less than half the events?

RE: Rephrased.

**15.** P191549 "contributing to ~30ppbv to surface ozone". I believe this is a modeling result. Please state this.

RE: Rephrased.

**16.** *P20 1555* "wildfire event". This would be a good place to summarize whether any of the sensitivity tests resulted in a better capture of the wildfire event.

**RE: Agreed. Please see below in the Conclusion section:**

"Although GFDL-AM3 captures the observed interannual variability in O3 enhancements from large-scale wildfires over the WUS (Lin et al., 2017), GFDL-AM4 has difficulty simulating the observed O3 enhancements during the relatively small-scale wildfire event on June 22. Sensitivity simulations with fire emissions constrained at the surface or with part of fire NOx emissions emitted as PAN do not substantially improve simulated O3 on June 22. Wildfires typically occur under hot and dry conditions which also enable the buildup of O3 produced from regional anthropogenic emissions, complicating an unambiguous attribution of the high-O3 events solely based on observations. Screening of exceptional events due to wildfire emissions remains a serious challenge."

**17.** *P18, 1500 "likely reflect": what are the arguments for this? It might be interesting to show a difference map for USB, getting at the uncertainty in USB between two state-of-the-art models.*

**RE:** The difference map for USA is now shown in Fig.S12.

---

## Author Response (AR2)

**Dear Dr. West:**

**We thank the two reviewers for carefully reading the manuscript again and for recognizing the efforts we made in the revised manuscript. Below we include a point-by-point response (in blue) to the reviewers'** *additional comments***, and explaining the changes made to the manuscript.**

**Best regards,**
**Meiyun Lin (on behalf of the authors)**
**July 13, 2020**

**Reviewer #1**

The authors have been responsive to reviewer comments and the revised manuscript addresses many of my concerns. I have a few remaining suggestions for improvement to the article before publication.

Abstract: The authors added a lot of nuanced language and discussion about uncertainties to the main text of the paper, but this language did not make it into the abstract which still talks about "pinpointing" sources and appears to suggest definitive attribution of O3 events.
RE: We have slightly revised the abstract to reflect uncertainties. Please see tracked changes.

Section 4: I still think a bit more information about model performance would be useful here. I suggest that the authors add a table which includes the model MB for AM4 and GC at the LV surface sites on each of the specific episode days examined (i.e. have a separate row for each of the dates analyzed in section 4).
RE: Thank you for the comment. Note that the model performance of MDA8 $O_3$ on each of the specific episode days examined is already shown in Table 1 for Angel Peak or Spring Mountain Youth Camp – the two representative baseline sites for Clark County and in Supplemental Table S4 and Fig.4 for the LV surface sites. We have clarified this in the beginning of Section 4 of the revised manuscript:

"Observations and model simulations of MDA8 $O_3$ for each event are also included in Table 1 for Angel Peak and in Supplementary Table S4 and Fig.S4 for all Clark County surface sites."

Readers can easily obtain the information for the model biases from the first four columns of Table 1. In the interest of keeping the paper succedent, we would prefer not to add any additional tables or figures in the main text.

Minor comments:
Note that the term "exceptional events" has specific regulatory meaning and specifically results in exclusion of data from the regulatory process.

Figure S5 (Apr 23) looks like O3strat_anomoloyg is much lower (5-10 ppb) but this doesn't seem to match purple lines on Fig 4
Note that $O_3$Strat in this Figure and Fig.9 is shown as anomalies relative to the monthly mean, while the absolute values are shown in Figs.4 and 8. We have clarified this in the figure captures in the revised manuscript.

Page 13, second paragraph: Note that AM4 does not definitively show the strat intrusion reaching ground level.
RE: We did not specifically mention in the paragraph that the intrusion has reached the surface. But it is clear from Figure 9 that there were episodic enhancements in surface $O_3$Strat, indicating that the intrusion has reached the surface.

Page 21, end of second paragraph: This is a good place to note that these quantitative model attributions are only as good as the precision/capability of the model which is why model eval is so important. This will also transition well into the next paragraph.
**RE: DONE.**

Page 23: increasing surface O3 in China may not indicate increased long-range transport of ozone to the US if the higher O3 is due to less titration from decreased NOx emissions. Atmospheric chemistry theory predicts that even when O3 is locally suppressed from NOx, high O3 forms downwind. So, if O3 is increasing due to less suppression that does not mean that more total ozone will be created/transported downwind.
**RE: Thanks for your insight. We have deleted ", contributing to increased background ozone" in that sentence.**

Figure 6: r2 value appears to be missing from the panel showing June 22.
RE: Nice catch! Corrected.

Reviewer #2

The revised version of this manuscript has resolved most of my questions with regard to this paper. I would recommend publication after the authors clarify the details below (all line numbers refer to the marked version at the end of the comments to reviewers). Most of these are rather minor although I am concerned that AM4 might be somewhat overzealous in its STT. This is addressed in comments (1) and (9). It would be good if the authors addressed this point, perhaps in the conclusions.

1. l349 of revised document. It is unclear how the STT events in Figure 4 are defined. On approximately May 7 for example the AM4 shows a peak in stratospheric ozone at SM Youth Camp with elevated ozone (not seen in the observations) which appears to be a STT event in the model. However, this event is not marked. There are other instances when the model seems to show high stratospheric ozone values that are not marked as STT events. It appears that the AM4 may show a number of false positives? Please explain and clarify. Also please comment with regards to these so called "false positives". (One of the major points of the paper is the importance of the ability of AM4 to simulate STT events. However, it looks to me like it may actually simulate too many of these events.)
Re: We made the following changes in the revised manuscript to address the reviewer' comment:

1.   In the caption of Figure 4, we now clarify "The blue shading highlights the STT events when observed MDA8 O$_3$ and AM4 O$_3$Strat show concurrent peak enhancements".

2.   When discussing the results in Fig.4, we noted:

"For some days, GFDL-AM4 overestimates total MDA8 O$_3$ due to excessive STT influence (e.g., May 7 at Spring Mountain Youth Camp)."

3.   In the Conclusions, we noted:

AM3/AM4 typically spreads the STT enhancement across a wider range of sites over the Southwest rather than capturing the observed localized feature, as discussed in more details by Lin et al. (JGR2012).  Thus, we propose targeted analysis of the observed high-O$_3$ events, rather than the modeled events, and recommend bias correction to simulated USB O$_3$ in AM4, such as the approach used by Lin et al. (2012a).

2. Figure S4, Figure 9 and analogous figures. Please give the spatial correlation between the simulated and measured ozone for both models? And perhaps the mean bias. It looks like AM4 is does not capture the geographic anomaly pattern. In fact in a number of locations in seems like the USB in AM4 is higher than that measured.

RE: The mean biases and spatial correlations between observations and model simulations of MDA8 O3 at CASTNet sites are now reported in Supplemental Table S4. Due to the sparse spatial coverage of CASTNet data, we believe that the spatial correlation coefficient is NOT a good measure of model's ability to represent the geographic anomaly pattern driven by deep STT. While AQS data have a vast spatial coverage, the variability across AQS sites may reflect the influence from processes other than STT, such as urban-to-rural chemical regimes.

As mentioned earlier, AM4 typically spreads the STT enhancement across a wider range of sites rather than capturing the observed localized feature, leading to low spatial correlations with CASTNet data. Taking the May 13 event as an example (Fig.S6), AM4 captures the observed strong MDA8 O$_3$ enhancements due to STT influence over Southern Nevada but spreads the STT influence too wide spatially. The observed O$_3$ enhancement over S. Nevada is completely absent in GEOS-Chem despite GC gives a higher spatial correlation with CASTNET data.

For these reasons, we keep these statistics in Table S4 and do not report them directly on Fig.9 and analogous figures. We think these numbers are not that useful in terms of gauging model's ability in representing STT.

3. L537 in track change manuscript. But in fact, despite the similarity in structure on an isentropic surface the two patterns in ozone, and its relations to isentropes still look remarkably different. Where the ozone is, the shape of the 322 isentrope with respect to the fold, the shape of the tropopause etc.

4. Figure 7 and discussion. The pattern of isentropic surfaces between the two simulations still look significantly different. For example, looking at the 322 K isentropic surface: at the southern edge of the slice this surface is significantly lower than in AM4 than in GEOS-chem; in AM4 it goes right through the fold whereas in GEOS-chem it is above the fold; north of the fold it is located higher in AM4 than in GEOS-chem and then descends near the north boundary in AM4 whereas in GEOS-chem it remains level. This difference really suggests a remarkable difference in the two meteorological analysis over a data rich region. It seems this should be pointed out. It is unclear to the extent to which the model differences are caused by difference in the meteorology. The difference in the 322 K contour may explain, in part, some of the differences in ozone between the two simulations. It is certainly worthwhile pointing out the large differences between the simulations south of the ozonesonde profile also. These differences are large and seem significant.

RE: This is now briefly discussed in the revised manuscript:
**"There are also some notable differences in the isentropic surfaces (e.g., at 322 K) between the two**

**models, possibly resulting from a difference in the two meteorological reanalysis data (NCEP in**

**AM4 and MERRA in GEOS-Chem). "**

5. l675. "likely due to excessive O3". I understand that in general lightning NOx is too high in the GEOS-chem simulations, but what is the evidence that on this particular day (June 22) it is likely the excessive ozone is due to lightning NOx. Were there thunderstorms in the vicinity, or …. ? "likely" seems a strong word without some in depth analysis.
RE: We now made this discussion more general, not just limited to the June 22 event, in the revised manuscript:

**"GEOS-Chem overestimates of free tropospheric ozone seem to be common for the non-STT events from late spring through summer (Figs.3b; Fig.8b, Fig.11b, and comparisons with lidar data on May 24 and June 16 in Sect. 4.5 and 4.6), likely due to excessive O$_3$ produced from lightning NO$_x$ over the southern U.S. (Zhang et al., 2011; Zhang et al., 2014). "**

6. Figure 9 and similar plots. I may have missed this, but why are some of the observed measurements shown in large dots, other in small dots?

RE: Small squares are for AQS observations and large circles are for CASTNet observations. This is now clarified in the figure caption.

7. I think the paper did not comment on **Figure 16b**. All figures show should certainly be commented on in the text. In fact it is quite remarkable that USB has a different North-South gradient in the two models. I don't believe this can be solely attributed to lightning NOx. Maybe I missed it, but is USB calculated the same way in both simulations. This figure is certainly worth a comment.

RE: USB is calculated in the same way in both models as described in Section 2. We believe the north-south gradient reflect differences in STT plus lighting NOx as discussed in the manuscript. Discussions in Section 5 are slightly revised and both Figures 16b and S12 are referenced.

8. Figure S12 was never comment on (again unless I missed it) in the main manuscript. It should be.

Re: Fig.S12 is now referenced in Section 5, along with discussions of Fig.16.

9. Figure S14. While it is true that the highest observed points show high values of stratospheric ozone it should be emphasized that for the most part those points in which AM4 are biased the stratospheric ozone is high. This seems like an important point to bring up. At least to me it seems quite likely that the AM4 model has too much STT (point 1). I think the authors should comment on this possibility.

RE: Thanks. We add this sentence to the revised manuscript:

[revised manuscript text omitted]